



# Characterization of Free Tropospheric Layers With Polar Radio Occultation Data

Terence L. Kubar[1,2], Manuel de la Torre Juárez[2], Jonas Katona[2,3], F. Joseph Turk[2]

[1]Joint Institute for Regional Earth System Science & Engineering, University of California, Los Angeles, 90095-7228, United States
[2]Jet Propulsion Laboratory, Pasadena, 91109, United States
[3]Applied Mathematics Program, Yale University, New Haven, 06511, United States

*Correspondence to*: Terence L. Kubar (tkubar@ucla.edu)

**Abstract.** Polarimetric Radio Occultation (PRO) concurrently detects heavy precipitation, ice, and the vertical thermodynamic structure inside clouds, enhancing traditional radio occultation measurements. We compare cloud top heights (CTOP) defined as the uppermost altitude at which the polarimetric phase difference between the horizontal and vertical components, $\Delta\phi$, exceeds 1 mm, for three years of PRO data from the Radio Occultation and Heavy Precipitation (ROHP) experiment, to the local tropical tropopause layer (TTL) base, a minimum stability level determined as the maximum lapse rate height (LRMAX). The TTL base coincides with the 80th - 90th percentiles of CTOP globally. We examine the skill of using the Tropopause Inversion Layer and the minimum vertical gradient of the lapse rate $(\partial LR/\partial z)_{min}$ to characterize the tropopause compared to the lapse rate and the cold point tropopauses. The TTL thickness, defined as the height of the $(\partial LR/\partial z)_{min}$ minus the LRMAX, is thinnest over the Tropical Warm Pool where LRMAX and CTOP are deepest. The steepest meridional gradient with latitude of the TTL top height is just equatorward of the subtropical maxima of the frequency of double tropopauses. For tropical raining clouds, when the maximum $\Delta\phi$, $\Delta\phi_{max}$ exceeds 10 mm, the mean binned CTOP is 2.7 km below the mean LRMAX, with a slope of nearly one. Using 0.8 mm for the $\Delta\phi$ CTOP threshold is optimal, while reducing below 0.8 mm decreases the CTOP and LRMAX spatial correlation. Globally, cloud tops associated with the largest 99th-percentile $\Delta\phi_{max}$ are 0.4 km above LRMAX.



## 1 Introduction

The troposphere can be subdivided into several layers. The lowest 1-2 km of the atmosphere is the Planetary Boundary Layer (PBL) where the atmosphere mostly exchanges energy and mass with the surface, and has been well characterized with Radio Occultation (RO) (e.g. von Engeln et al., 2005; Kalmus et al., 2022 and references therein). The region immediately above the PBL, the free troposphere, is unstable to convection and less influenced by Earth's surface. Above the troposphere is the stratosphere, where the atmosphere becomes stably stratified, heating is mostly through $O_3$ radiative absorption of direct solar

radiation, and convective activity rarely penetrates.

The height at which convective activity ends is typically marked by the top of convective clouds. Above clouds, water vapor mixing ratios become low enough to enable CO, $CO_2$ and $O_3$ to dominate radiative heating (e.g. Gettelman and Forster, 2002; SPARC, 2022). Fueglistaler et al. (2009) show that the upper limit to tropical convective activity can occasionally reach the cold point tropopause (CPT), typically defined as the coldest point of the two layers, the troposphere and stratosphere. The

upper tropical troposphere layer between the top of deep convection and the CPT shares properties with the stratosphere and has therefore also been called the substratosphere (Thuburn and Craig, 2002) or the Tropical Tropopause Layer (TTL) (see review in Fueglistaler et al., 2009). In this paper, we use both substratosphere and TTL equivalently. In the substratosphere, the timescale of convective transport is slower than the radiative timescale, yet $O_3$ is still low due to occasional deep convection penetration.

Boundaries of the TTL have been characterized using RO in Schmidt et al. (2004) and Sunilkumar et al. (2017) using thermal stability criteria. While the CPT is often defined as the upper bound to the TTL, Sunilkumar et al. (2017) alternatively use the level of maximum stability, defined by the minimum lapse rate, as the top boundary of the TTL. $O_3$ radiative heating often causes a sharp thermal inversion above the CPT that can cause this maximum stability level. This inversion layer separating the CPT and the level of maximum stability is the Tropopause Inversion Layer (TIL) (Birner et al., 2002; Randel et al., 2007).

Therefore, in our study, we explore the use of the TIL captured by the height at which the minimum lapse rate occurs (LRMIN), as one of several tropopause metrics compared, which may be an even tighter upper boundary of tropospheric air than the CPT.

The CPT occasionally acts as a strong limit of uppermost cloud tops, but the TTL base, several km below especially in the tropics, has also been associated with the top of deep convection (Thuburn and Craig, 2002). The maximum lapse rate height

(LRMAX) in the free troposphere may set the effective height of maximum cloud detrainment below which temperatures are convectively adjusted (Fueglistaler et al., 2009; Chae and Sherwood, 2010). The single-column radiative-convective models from Thuburn and Craig (2002) show that the lapse rate near convective cloud tops drives heating locally with $CO_2$ near 155 hPa, thereby stabilizing the layer between the top of convection and the CPT. Early measurements of $CO_2$ from commercial aircraft by Georgii and Jost (1969) demonstrated that $CO_2$ is markedly higher in the upper troposphere by 3-5 ppmv versus the

lower stratosphere - the jump in $CO_2$ was considered indicative of the CPT. Sensitivity experiments by Thuburn and Craig (2002) reveal that $CO_2$ concentrations of approximately 0.3% of contemporary values or less would nearly completely remove





the separation between convection and the CPT, the substratosphere, whereas increasing $CO_2$ beyond current values is predicted to increase the thickness of the tropical substratosphere above today's climatological depth. Water vapor has a strong peak in net cooling near or just below 200 hPa, near the top of deep convection (Thuburn and Craig, 2002). This strong peak

can move convection and the CPT higher. Understanding what drives the altitude of clouds and the CPT, the depth of the TTL, and their response to a changing climate make the high vertical resolution of all-weather RO observations and the sensitivity of polarimetric RO (PRO) to ice (Padullés et al., 2023) good tools for quantifying subtle changes in stability and layers of deep convective clouds.

The CPT marks the transition to stratospheric circulation patterns, like an increasing dominance of $O_3$ heating (Thuburn and

Craig, 2002), the Brewer-Dobson circulation (BDC) (Fueglistaler et al., 2009), and the Quasi-Biennial Oscillation (QBO). However, some stratospheric phenomena can create a minimum temperature well within the stratosphere, making the CPT definition fail as marker of the tropopause-e.g., during polar night when the lack of solar heating leads to a cooling of the stratosphere, thereby placing the coldest altitude between 20 km and 30 km. As also noted in Thuburn and Craig (2002), the WMO lapse rate criterion may occasionally not be satisfied below ~20 km. However, even in polar night, there is still a remnant

of a TIL marking the transition between $H_2O$ radiative cooling and $O_3$ heating. We use this to explore a modified criterion to find the CPT based on sharpness in the vertical change of the temperature lapse rate.

We note that other attempts also exist in the literature to overcome challenges associated with the CPT not always representing the real tropopause, especially away from the tropics (e.g. Munchak and Pan, 2014). Xia et al. (2021) use refractivity profiles to identify the tropopause height. Their method relies on using the wavelet covariance transform

(Gamage and Hagelberg, 1993) to estimate where the steepest change in refractivity occurs, which the authors also indicate marks the transition in the vertical from steep to reduced vertical temperature gradients.

The WMO lapse rate tropopause, which we briefly assess, often approaches our proposed sharpness-based CPT with modified criterion, but it can sometimes be found in the mid-troposphere. Some stratospheric intrusion events may cause two concurrent tropopauses, where the bottom layer may correspond to cloud tops, while the upper tropopause is linked to the

poleward advection of tropical upper tropospheric air. In the tropics, according to the work by Biondi et al. (2011), the dynamics behind double tropopauses over low-latitudes may be more driven by deep convection, and can only be more robustly observed using co-located radiosonde data.

Both RO and PRO capture the transition altitudes between the tropospheric layers, separating the PBL from the free troposphere (e.g. Kalmus et al., 2022), the free troposphere and free-tropospheric cloud tops from the TTL (e.g., Peng et al.,

2006), and the TTL top (tropopause) from the stratosphere (Randel et al., 2007). This study focuses on the upper layers. We briefly touch upon WMO tropopause detection before focusing on detecting the bottom of the TTL as defined by LRMAX, the tropopause based on the sharpness of the LR profile, and compare different metrics of defining the tropopause. Section 2 describes the Methods/Data used to conduct this analysis. Section 3 presents our results, and we divide into subsections as follows: in Sect. 3.1, we introduce a mid-latitude case study illustrating a double tropopause following the passage of an upper-

level trough, justifying our choice in Sect. 3.2 to calculate a surrogate for the CPT, valid at polar night latitudes, and presents





global statistics of TTL metrics and different measures of the tropopause. It includes how the distance between the modified CPT and coldest point in the stratosphere changes with latitude and season and statistical insights between the maximum absolute gradient change of LRMIN and the frequency of double tropopauses. Section 3.3 presents challenges of gravity waves in determination of the tropopause, showing how all tropopause metrics fail except for one. Section 3.4 characterizes the

seasonal global CTOP versus LRMAX relationships from both RO and PRO data. Section 3.5 presents maps of low-latitude distributions of CTOP, LRMAX, the temperature of LRMIN as an alternative proxy of the CPT, and the TTL Thickness. Section 3.6 assesses the sensitivity of different thresholds for CTOP in the tropics and the relationships with LRMAX. We explore how changes in the threshold affect the correlation between CTOP and LRMAX. Section 3.7 compares global and regional CTOP-LRMAX offset versus $[\Delta\phi]_{max}$, and presents a useful parameter in predicting the probability of heavy

precipitation. Section 4 summarizes the results and discusses the implications.

## 2 Methods and Data

### 2.1 Conventional RO data sources for upper tropospheric/tropopause variable definitions

To classify global statistics of the upper troposphere and tropopause metrics, conventional RO is used from Spire-RO (UCAR) (Spire, 2021) and PRO from the PAZ satellite, the latter of which was launched on 22 February 2018 (Cardellach et

al., 2019), and for which the data available for this study were taken from 26 July 2018 through 6 October 2021. Conventional PAZ data from January 2023 are also used for a California case study illustrating how we deal with a double tropopause. Refractivity, $N(z)$, is the primary variable retrieved from conventional RO profiles, from which temperature and moisture can be retrieved using the following equation (e.g. Kursinski et al., 1997):

$$N(z) = 77.6\frac{p(z)}{T(z)} + 3.73x10^5\frac{e(z)}{T^2(z)} + 4.03x10^7\frac{n_e}{f^2} + 1.4W, \quad (1)$$

where $p$ is pressure in hectopascals (hPa), $T$ is temperature in degrees Kelvin, $e$ is the water vapor partial pressure in hPa, $n_e$ is the electron number density per cubic meter, $f$ is transmitter frequency in Hz, and W is the liquid water content in g/m$^3$. The first two terms of Eq. (1) are the dry and wet terms, respectively, (e.g. Smith and Weintraub, 1953), the third term is the ionospheric term, and the fourth term is either the scattering term (Kursinski et al., 1997) or simply the liquid water content (LWC) term and contributes a small fraction of the other terms (Yang and Zou, 2012). Typically, the ionospheric term is

removed during Global Navigation Satellite System (GNSS) retrievals.

$T$ and $p$ are derived from $N$ using hydrostatic balance in the upper atmosphere where the contribution of $e$ is almost zero. In the lower and middle troposphere, below the Tropical Tropopause Layer (TTL), $e$ becomes important. The standard approach follows Kursinski et al. (1997), and uses ancillary temperature information for mid-tropospheric heights where T>250K. Most of the layers studied here are above this height and the results will be unaffected by the uncertainties associated

with not knowing $e$. Both Kursinski et al. (1995) and Mascio et al. (2021) state that RO constrains water vapor below the 250-K isotherm and temperature above the 250-K isotherm. Minimal impacts have been reported on using RO to estimate TTL



height estimates in the tropics and subtropics by Kursinski et al. (2000); estimates for a saturated atmosphere indicate that for $p < 250$ hPa and $T < 230$K, the wet term of (1) is less than 0.5% of total $N$. At lower altitudes (higher pressures), the temperature threshold for $e$ to exceed 0.5% increases further.

We only search for the TTL with a base colder than 250K, because above this altitude RO use dry retrievals as the standard practice and change to using model temperatures at lower altitudes. This criterion eliminates changes in the LR caused by the switch to model temperatures (Kursinski et al., 1995; Padullés et al., 2020), and discards profiles where the lapse rate changes below the altitude where 250 K is crossed. This bias doesn't affect the relationship between where the TTL base and CTOP are. To minimize influences from PBL processes, we start searching for the TTL base above 6 km (e.g. Kubar et al. (2024)

used 0-6 km to look for PBL heights and related clouds). We address specifically whether the height of the maximum lapse rate, henceforth referred to as LRMAX, which represents the minimum stability and has been proposed as a definition of the TTL base (Chae and Sherwood, 2010), can appropriately mark the top of non-penetrating convective clouds. We set an upper bound of 20 km in which to look for LRMAX.

Previous work has classified the CPT as part of the upper troposphere (e.g. Fueglistaler et al., 2009), but during polar winter

the actual coldest temperature is often found in the mid-stratosphere, presenting challenges in using the coldest point as a sole tropopause definition. The WMO definition of the tropopause, the lowermost height at which an LR threshold of 2°C/km is sustained for at least 2 km, often captures the real tropopause, though when relatively shallow clouds are capped by a strong and deep inversion, that inversion may be misconstrued as the tropopause and instead represent the cloud top height (Biondi et al., 2012; Peng et al., 2006). We assign the height of the modified CPT criterion to where the vertical gradient of the lapse

rate profile reaches a minimum, $(\partial LR/\partial z)_{min.}$ The second derivative of the temperature profile captures the heights with the sharpest change in the sign of the lapse rate.

## 2.2 Polarimetric RO data from ROHP experiment aboard the PAZ satellite

CTOP is defined in PRO profiles as the uppermost layer at which the differential phase shift ($\Delta\phi$) exceeds a given threshold; we estimate CTOP from the Radio Occultation and Heavy Precipitation (ROHP) experiment and draw from a possible 151,823

profiles from July 2018 through October 2021. Joint CTOP-LRMAX offset versus $[\Delta\phi]_{max}$ histograms are generated using PRO and IMERG/GPM brightness temperature data in order to validate PRO data with independent cloud top satellite data products. IMERG in NASA's Integrated Multi-satellitE Retrievals for the Global Precipitation Mission (GPM). It is an algorithm combining information from the GPM satellite constellation to estimate precipitation over the Earth's surface. In Sect. 3.5, we evaluate the sensitivity of the $\Delta\phi$ threshold by examining binned relationships over the low latitudes of CTOP

versus LRMAX, and over the globe in Sect. 3.6.

## 3  Results



### 3.1 Mid-latitude double tropopause case study

We begin by presenting an example of a profile of temperature with conventional PAZ RO data which has complex layering

characteristic of a double-tropopause case fairly common during the winter in the subtropical-to-mid-latitudes, and in doing so also visually present the different tropopause definitions utilized in our study.

Figure 1 presents conventional RO profiles from the PAZ satellite at 37.3°N, 120°W, well east of Oakland, CA at 17:29 LT on 31 January 2023. This case was manually chosen originally from pre-existing radiosonde data (not analyzed here) in light of a significant separation between the height of $(\partial LR/\partial z)_{min}$ and the CPT. Inspection of many profiles in the mid-latitudes

during winter, especially in conjunction with the passage of upper-level low pressure systems, reveals that such profiles are not all that uncommon. In panel (a), the original PAZ data (of ~100 m native vertical resolution) are regridded onto equal 0.1 km vertical grids and also filtered with a Single Spectrum Analysis (SSA) (Ghil et al., 2002) to capture the most salient features of the temperature profile. SSA provides the eigenvalue and eigenvector spectrum of a principal component analysis of the signal variability, and we use it here as a low-pass filter technique to extract the larger variability structures. The temperature

profile shown is effectively the dry temperature, e.g. where the contribution from water vapor becomes small enough such that term (2) in Eq. (1) contributes negligibly, compared to term (1), which happens when T < 250 K (Kursinski et al., 1997), as discussed in the Methods and Data section. In Fig. 1, this occurs for heights above about 5 km. The strong decrease of temperature with height, characteristic of the free troposphere, continues up to about 10.5 km, also readily seen by the LR profile (Fig. 1b). An exception to this is around 8.5 km- 9 km where a stable layer exists, also apparent when examining the

LR profile constructed using regridded retrieved temperature at 0.5 km vertical resolution.

Figure 1b presents all the relevant layers, including the WMO tropopause, which is the first layer above which the LR < 2°C/km for at least 2 km above, which here is 10.25 km. Our modified CPT criterion is the height corresponding to $(\partial LR/\partial z)_{min}$, and is just a tad higher, 10.40 km, and LRMIN is only 350 m higher than that.

We considered if the deep layer between the two tropopauses could be TTL-like, above the first tropopause is a stable layer

with a small negative LR, between about 10.25 km and 13.5km; but in the tropics above the LRMAX, while LR decreases with height it still is strongly positive (Sunilkumar et al., 2017). Above that slightly stratified stable Oakland, stability decreases once again, extending up to the actual CPT. Near-neutral conditions are approached near 15.5 km, and though we do not have PRO data available for this date, this decreased stability just below the CPT might be more conducive for the maintenance of possible thin ice clouds (cirrus). The tropical CPT, often around 16-17 km, frequently extends well beyond the low-latitudes,

and a vertical layer of decreased stability in a layer just below the CPT is often present, as in Fig. 1b.

In Fig. 1c, we show the vertical profile of $\partial LR/\partial z$ from both the binned 0.5-km resolution LR data and also the 0.1-km binned SSA LR profile, and clearly see how the modified CPT criterion based on sharpness stands out around 10.25 km. The CPT generally coincides with the second strongest critical value of $\partial LR/\partial z$ in terms of a *local minimum*.




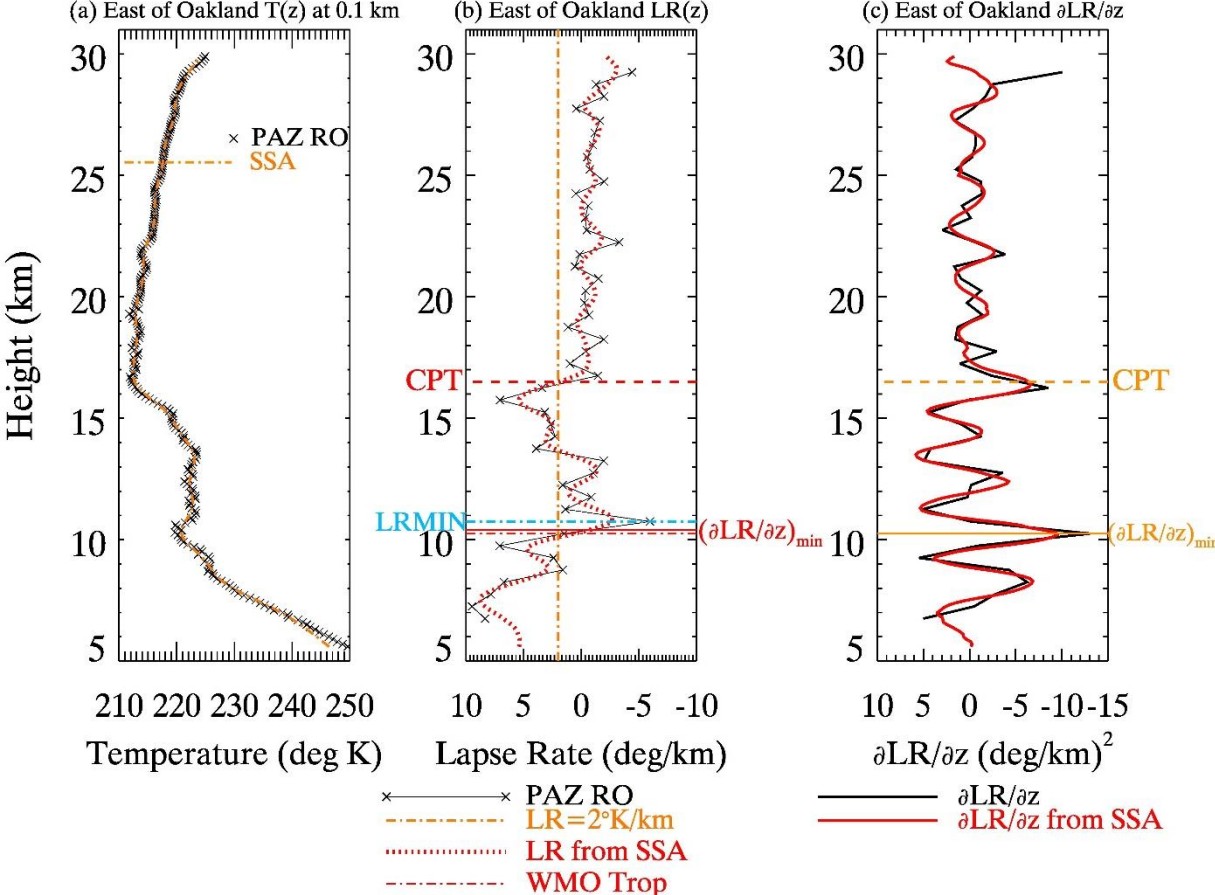

**Figure 1:** (a) Conventional RO profiles from PAZ of temperatures from 30 January 2023 at 17:29 Local Time (LT) at 37.3°N, 120°W, east of Oakland, CA by about 2.2° with a binned vertical resolution of 0.1 km. Also included is the singular spectrum analysis fit (orange dashed-dot-dashed line). (b) Lapse rate ($-\partial T/\partial z$) profile using 0.5-km vertical resolution temperature retrievals, and also the LR from the SSA at 0.2 km resolution (red short dashed). The LR=2°/km line is shown, along with the estimated WMO tropopause (10.25 km), the height of the sharpest temperature bend, min($\partial LR/\partial z$) (10.40 km), height of LRMIN (10.75 km), and the CPT (16.50 km), based on the 0.5 km resolution LR/T profiles. Note that the LR in the upper TTL, or just below the CPT, reaches near-neutral conditions. (c) $\partial LR/\partial z$ profiles calculated from the binned LR profiles (black) and from using the LR from the SSA profile in panel (b) (red). Shown for reference are both the CPT and the modified criterion for the CPT, red in (b) and orange in (c).

## 3.2 Global statistics of relevant tropospheric and lower stratospheric layers

We next highlight global statistics from Spire-RO data of several of the aforementioned layers of the troposphere, TTL, and a few aforementioned tropopause metrics from two periods in which polar night is approached to demonstrate areas of agreement between the standard CPT, based on temperature, and our modified criterion for the CPT, based on LR sharpness, as a function of latitude, and where separation is greatest. The top (bottom) panels of Fig. 2 show data from 1-5 December 2022 (1-5 June 2023), respectively, when the northern (southern) hemisphere high-latitude regions are in or approaching polar night. In these panels we show not only data from individual profiles but also data averaged into 4-degree latitude bins for additional clarity.



Small black (red) dots show each individual CPT (modified CPT) versus latitude. Over the low-latitudes, especially |LAT|<20°, the modified CPT definition versus the standard CPT are very close to each other, within about 100-200m, the typical vertical

resolution of RO profiles. We also present median WMO-LR tropopause heights in 4-degree latitude bins, and while they are quite similar as well to our modified CPT, they are a bit shallower over the tropics, which we zoom in on momentarily. The median CPT diverges substantially from the sharpness-defined CPT poleward of the tropics, with the modified CPT decreasing most quickly with latitude around 25° latitude, regardless of season and hemisphere, though with a sharper decrease at that latitude in the southern hemisphere. Poleward of about 30°, two branches of the CPT (black symbols) develop, one lower one

closer to the sharpness-based CPT and one an upper CPT, either reflective of a second tropopause or the TTL top that extends to the subtropics. Not far from the emergence of these two branches is where double tropopauses are most common statistically, which we show in Fig. 3 and also overlay with $|\partial(\partial LR/\partial z)_{min}/\partial LAT|$. The strongest latitudinal gradient of $(\partial LR/\partial z)_{min}$ maximizes just equatorward of the double tropopause frequency maxima, especially in the Northern Hemisphere; however, both the maximum latitudinal gradient of $(\partial LR/\partial z)_{min}$ and the likelihood of double tropopauses are greater in the subtropical

southern hemisphere.

Homing in on lower latitudes (|LAT|<35°), we see even more clearly that the sharpness-based CPT is close to the standard CPT, within ~150 m, for |LAT|<20° during both December 2022 (Fig. 4a) and June 2023 (Fig. 4b), whereas the WMO-LR tropopause is about 300-600 m shallower during either month. This suggests that $(\partial LR/\partial z)_{min}$ may better capture the height of the bend of the temperature profile from decreasing with height (tropical troposphere) to increasing with height (stratosphere)

particularly in the tropics than the lowermost height at which LR drops below 2°C/km. Figure 4 also demonstrates that both the sharpness-based CPT and the WMO-LR tropopause decrease precipitously with latitude particularly around LAT ~ |25°|, whereas the CPT retains its high tropical values of ~17.0-17.5 km well poleward of there. We also note that, especially over the Southern Hemisphere, that on average the height of $(\partial LR/\partial z)_{min}$ from about ~25°S-30°S is slightly lower than the WMO tropopause; this occasionally occurs when a relatively shallow cloud top is associated with or leaves a strong imprint in the

LR signature, which may be more common in this region. Occasionally there are also some instances in which the maximum negative sharpness of the LR is just below the height at which the LR drops to 2°C/km.

Returning to Fig. 2a, during the 1-5 December 2022 period, the highest CPT has median values between 28km-29km north of 60°N; this is not apparent during austral early meteorological summer. During 1-5 June 2023, the deepest CPT is found in the high latitudes of the southern hemisphere, albeit at slightly lower altitudes than the northern hemisphere, with a peak just

poleward of 50°S.

As polar night approaches and sunlight is either minimal or nil, there is little to no absorption of UV-radiation by stratospheric ozone, allowing the coldest absolute temperatures to often form well into the lower- to mid-stratosphere. Clearly these are not tropopauses that would separate two atmospheric layers; this occasionally happens with our modified CPT criterion as well, albeit less so during June, but we note that this happens less frequently with the WMO definition of the

tropopause, though statistically, particularly away from the tropics, the differences between the WMO LR tropopause and the modified CPT are small.



To further corroborate that ozone is an important contributor to lower stratospheric heating, we also include LRMIN in both panels of Fig. 2, which generally is about 1.5-2.0 km above the standard CPT or the modified CPT in the tropics and about 2 km above the modified CPT in the polar night region, where there is little sunlight, but in the southern hemisphere during December, the highest stability (LRMIN) approaches 18 km near the South Pole where sunlight is abundant, which may be a reflection of ozone radiative heating at those levels. We also note that LRMIN is also frequently above 20 km in the subtropics and mid-latitudes, but we exclude those cases here as we present binned median LRMIN versus latitude for observations between 6 km and 20 km.

Both panels in Fig. 2 both also plot LRMAX, which has been proposed as the base of the TTL, and we will also show momentarily with PRO data to coincide with the local deepest clouds. LRMAX is highest near the equator at around 12 km, also coincidentally the typical detrainment height of tropical anvil clouds (Chae and Sherwood, 2010; Kubar and Jiang, 2019), coinciding with cloud top temperatures between 220K-225K (Kubar et al., 2007). During 1-5 June, higher LRMAX is apparent just north of the equator, perhaps a reflection of stronger convection during the boreal early summer and greater land fraction just to the north versus just to the south of the equator.

Even though PAZ P-RO data are not coincident with Spire-RO data, CTOP from PAZ $\Delta\phi$ profiles provide additional context about LRMAX as a potential upper limit for most cloud tops, and thus an appropriate metric of the TTL not only in the tropics, where it is traditionally defined and found, but also in the mid- and high-latitudes as a rough correlate with cloud detrainment. Figure 2 shows the 75th and 90th percentiles of CTOP ranked by height for each 6° latitude bin, which are the statistically deeper clouds in a region which would be more likely to be connected to LRMAX, and also this further separates these clouds from possible pure PBL-topped clouds, which are ubiquitous throughout the globe.

A few additional remarks emerge from showing CTOP: in the tropics, the 75th/90th percentiles of CTOP are often at or below LRMAX, whereas LRMAX appears to fall somewhere between the 75th/90th CTOP percentiles over the mid- and high-latitudes. There is a seasonal shift of where the deepest clouds are – they tend to be over the summer hemisphere poleward of the equator – June for the Northern Hemisphere and December for the Southern Hemisphere. Also, LRMAX is clearly a maximum just north of the equator in June, but there is no clear peak in location of LRMAX during December.

Note that while we show December and June CTOP distributions, these occur during different years from the Spire-RO data and thus there may be some year-to-year differences as well.







**Figure 2:** (Top) Scatterplots using Spire RO profiles from 1-5 December 2022 of defining either the tropopause or the bottom of the Tropical Transition Layer (TTL). Small black dots represent the coldest level of the atmosphere, and small red dots the height of $(\partial LR/\partial z)_{min}$, which is more related to the WMO LR tropopause. Also shown are median values in 4° latitude bins of the cold point height (solid cyan line), median height of min($\partial LR/\partial z$) (thick dashed cyan), WMO LR tropopause (dashed-dot orange curve), median LRMAX and LRMIN, and 75th/90th percentiles in 6 ° latitude bins from PAZ of CTOP from 2018-2020 December. Bottom: Same as top, except for 1-5 June 2023 and 2018-2021 for June for PAZ.





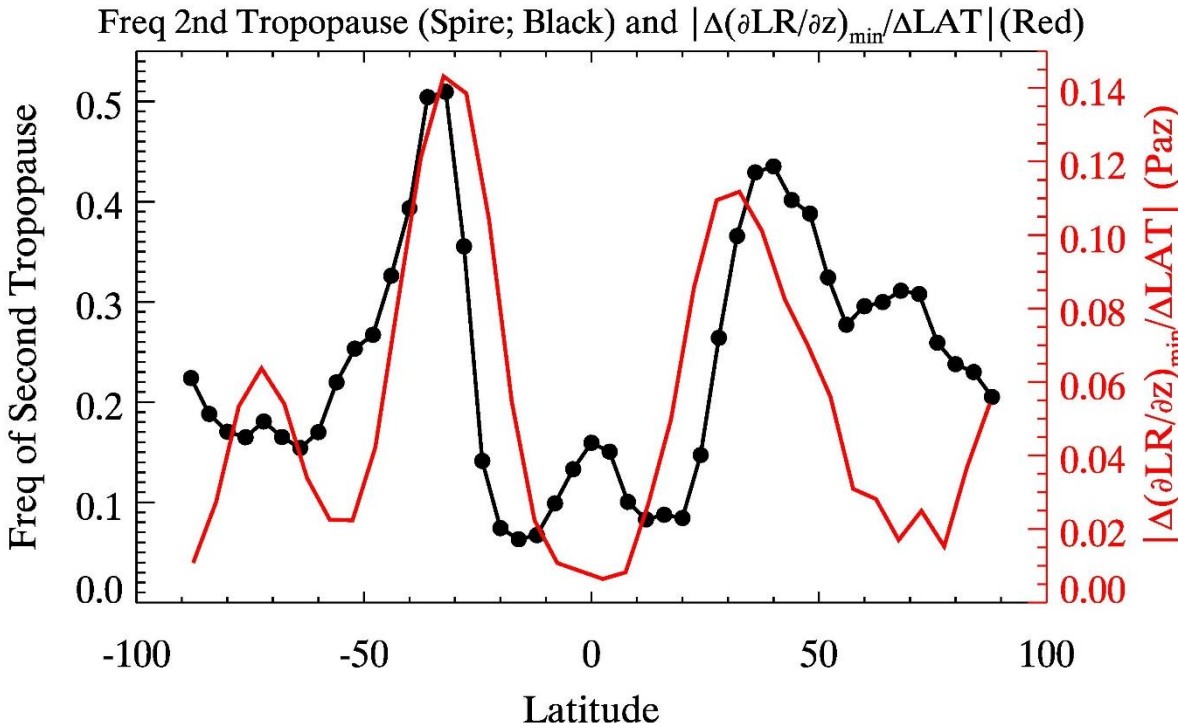

**Figure 3:** Frequency of two simultaneous tropopauses as a function of latitude, in four-degree latitude bins, from Spire RO data (black), and the derivative with respect to latitude of the height corresponding to the minimum vertical gradient of the lapse rate from Paz (red), which tends to coincide with the subtropical-to-mid-latitude double tropopause frequency peak, especially in the southern hemisphere.







**Figure 4:** CPT, modified CPT, and the WMO LR Tropopause versus latitude for |LAT|<35° in four-degree latitude increments for (a) December 2022 and (b) June 2023 in order to show the stronger coherence between CPT and our modified CPT especially equatorward of 20° latitude.



### 3.3 Gravity wave interference with determination of the tropopause

While we have thus far demonstrated the merits of a sharpness-based CPT which usually captures the standard tropical CPT
for latitudes < 25° and often the mid- and high-latitude WMO-tropopause poleward of about 30° latitude, we indicated cases
in which any of the tropopause definitions may indeed be outliers or perhaps inconsistent with representing the level which
physically separates tropospheric air from stratospheric air. This can occur when there is considerable stratosphere-to-
troposphere exchange of air masses, or when particularly strong gravity waves are generated, induced by a strong increase in

stability, particularly near and just above the tropopause (Fetzer and Gille, 1994; Fritts and Alexander, 2003; Miao et al., 2022;
Shao et al., 2023). Model temperature profiles and other filtering (e.g., SSA) techniques tend to smooth out vertical temperature
perturbations associated with gravity waves, and may also misplace the actual tropopause. We provide an example of such a
tropical case between Cuba and The Bahamas in Fig. 5 at 22.39°N, 76.89°W, with the left panel showing PAZ-RO and NCEP
model temperature profiles, and the right panel PAZ-NCEP T in which the WMO tropopause (short dash) of 14.5 km

appropriately captures the tropopause, whereas the calculated RO CPT (16.7 km) and RO $(\partial LR/\partial z)_{min}$ heights (16.8 km)
correspond to a temperature minimum/inversion well above the real tropopause. Even the NCEP temperature profile yields a
CPT of 15.6 km, still above the actual tropopause.

The height of the WMO tropopause marks a local cold perturbation in RO T – NCEP T (Fig. 5, right panel) and the LRMIN
from RO near 16.0 km marks the top of the TIL, coincident with maximum (RO T – NCEP T) of over 3°C.  Even LRMAX

near 12 km may be of interest as it shows a local minimum perturbation, likely marking the top of a cloud layer. The main
point of this exercise is that there are profiles in which the standard CPT and sharpness-based CPT are below the TIL but that
gravity waves perturb the shape of the tropopause, making it difficult to discern where the transition is from tropospheric to
stratospheric air.




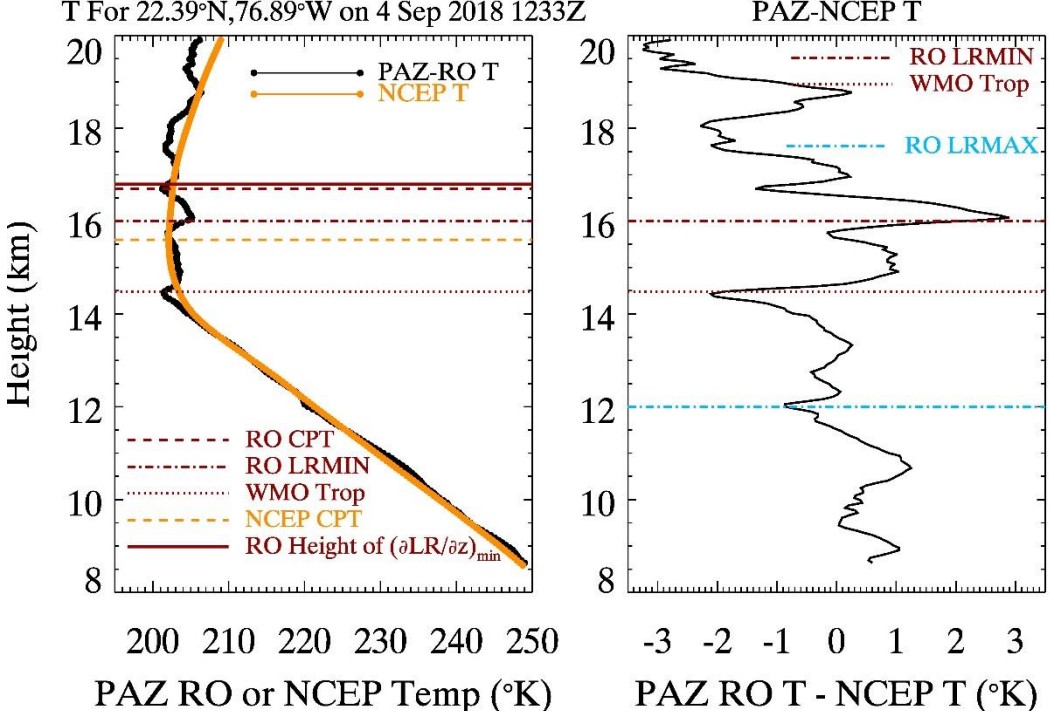

**Figure 5:** Example of a profile in which temperature perturbations associated with gravity waves interfere with proper identification of the standard CPT or sharpness-based CPT $(\partial LR/\partial z)_{min}$ and even how a smoothed profile (NCEP) mischaracterizes the standard CPT. (Left) Paz RO T and NCEP T profiles, with calculated WMO tropopause (from RO), RO CPT, RO Height of $(\partial LR/\partial z)_{min}$, RO LRMIN, and NCEP CPT. (Right) Paz Minus NCEP T with shown RO LRMAX, RO LMIN, and the WMO tropopause.

### 3.4 Seasonal global CTOP versus LRMAX relationships from Spire and PAZ

In Fig. 6, we extend the global PAZ versus LRMAX relationships, with the addition here from Fig. 2 that we present LRMAX from Paz, which is co-located with our estimates of CTOP based on the uppermost height at which $\Delta\phi > 1$mm. Again the $75^{th}/90^{th}$ percentiles of CTOP are ranked by height in each six-degree latitude bin. Over the mid- and high-latitudes, there are some differences in LRMAX from the different datasets, though some of this could be due to a slightly different averaging technique, different time periods, or even that we only calculated LRMAX for PAZ if there was a cloud in the profile (though we checked offline and LRMAX estimates were relatively insensitive to this). Notably, CTOP appears to be modestly better spatially correlated with Spire LRMAX than with PAZ LRMAX, with correlation coefficients provided on panels (c) through (f) on Fig. 6. For both LRMAX estimates, the $75^{th}$ percentile of CTOP is always below LRMAX, but closer to the one-to-one line or even slightly above for the $90^{th}$ percentile of CTOP, suggesting that perhaps 10% or fewer clouds make it above the





TTL base. Interestingly, in the summer hemisphere over the high-latitudes, cloud tops are more likely to reach levels above

LRMAX regardless of dataset.

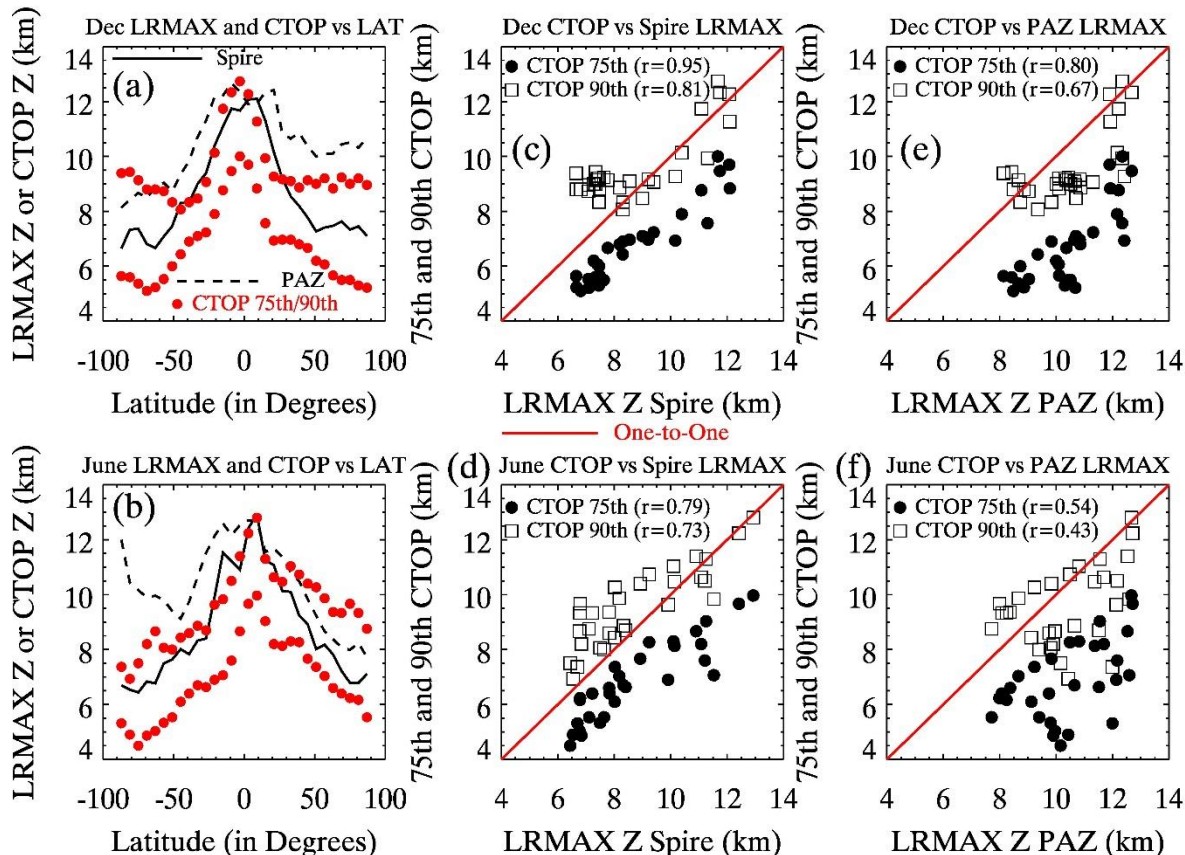

**Figure 6**: (a) LRMAX estimated from Spire for December 1st-5th, 2022 and from PAZ for December 2018, December 2019, and December 2020 longitudinally-averaged in six-degree latitude bins. Also shown are the corresponding 75th and 90th
percentiles (within each latitude bin) of CTOP from PAZ. (b) Same as (a), except for June 1st-5th, 2023 for Spire and June 2019, June 2020, and June 2021 for PAZ. (c) December 2018, December 2019, and December 2020 PAZ 75th/90th CTOP versus December 1st-5th, 20222 Spire LRMAX. (d) Same as (c) June 2019, June 2020, and June 2021 75th/90th CTOP vs Spire LRMAX for 1-5 June 2023. (e) Same as (c), except versus PAZ LRMAX for December 2018, December 2019, and December 2020. (f) Same as (d), except versus PAZ LRMAX for June 2019, June 2020, and June 2021. Red lines in panels (c) through
(f) indicate the one-to-one line. r-values are given in panels (c) through (f).

## 3.5  Low-latitude distributions of CTOP, LRMAX, LRMIN and TTL Thickness

A natural extension of the previous longitudinally-averaged analysis is to present maps of the distributions of the TTL base and height characteristics from PAZ-RO, and also the higher-percentile cloud tops from PAZ-PRO, with CTOP ranked by height in each 5° longitude x 5° latitude bin, which we examine in Fig. 7. Figure 7a presents the 75th to 90th percentiles of

CTOP, Fig. 6b mean LRMAX, Fig. 6c the temperatures corresponding to $(\partial LR/\partial z)_{min}$, and Fig. 7d presents a proxy for TTL



thickness, Z of $(\partial LR/\partial z)_{min}$ minus LRMAX, which is somewhat similar to Sunilkumar et al. (2017), though their upper boundary was the TIL. We opt for temperature of $(\partial LR/\partial z)_{min}$ since we have seen that it intrinsically resembles the tropical CPT, and tropopause heights have rather muted west-to-east tropical heterogeneity (not shown); in contrast, colder temperatures associated with the sharpness-based CPT are associated both with more upper-level instability (Emanuel et al.,
2013) and with different modes of the QBO and resultant formation of near-tropopause cirrus clouds (Son et al., 2017); this makes sense since the mean LRs in the tropical West Pacific and Indian Ocean are greater near the TTL and up to the either form of the tropical CPT.

The tropical warm pool over the western equatorial Pacific, Maritime Continent, and especially the eastern equatorial Indian Ocean is well-outlined by the deepest CTOP, with the highest clouds just west of and extending to the International date line
(e.g. 140°E-180°). While fairly deep clouds extend across the central and eastern Pacific ITCZ region north of the equator, they are shallower than the western Pacific warm pool; this is consistent with previous studies from MODIS (Kubar et al., 2007) or CloudSat (Kubar and Hartmann, 2008) demonstrating the gradient of cloud top height/temperature across the tropical west versus east Pacific. Figure 7b shows the distribution of mean LRMAX, and largely echoes CTOP, except over the geographical equator of the central and eastern Pacific in which a cold tongue of low SSTs exists, relative to the equatorial
western Pacific (Bjerknes, 1966), have relatively high heights despite a paucity of deep convection there. This suggests that LRMAX may be driven more by processes in the upper-troposphere, as discussed in the Introduction, namely a capturing of maximum longwave cooling by water vapor; LRMAX may be directly driven by clouds as well but is somewhat more spatially homogenous over shallow convection regions since the temperature at which clear-sky radiative cooling precipitously diminishes is steadier (Hartmann and Larson, 2002).

The temperature of $(\partial LR/\partial z)_{min}$ (Figure 7c) broadly resembles in an inverse way where the deepest convection is, particularly the coldest temperatures over the western tropical Pacific and Indian Ocean, with slightly warmer temperatures in the eastern equatorial Pacific where less convection is. Note that the latitudinal-width of very low temperatures is greater over the eastern hemisphere where deep convection is more prominent versus the eastern Pacific or even eastern Atlantic Ocean Basin.

Figure 7d shows the TTL thickness, Z of $(\partial LR/\partial z)_{min}$ – LRMAX; areas with the smallest TTL thickness represent deeper
convection, fairly consistent with Sunilkumar et al. (2017) and our CTOP map in panel (a).



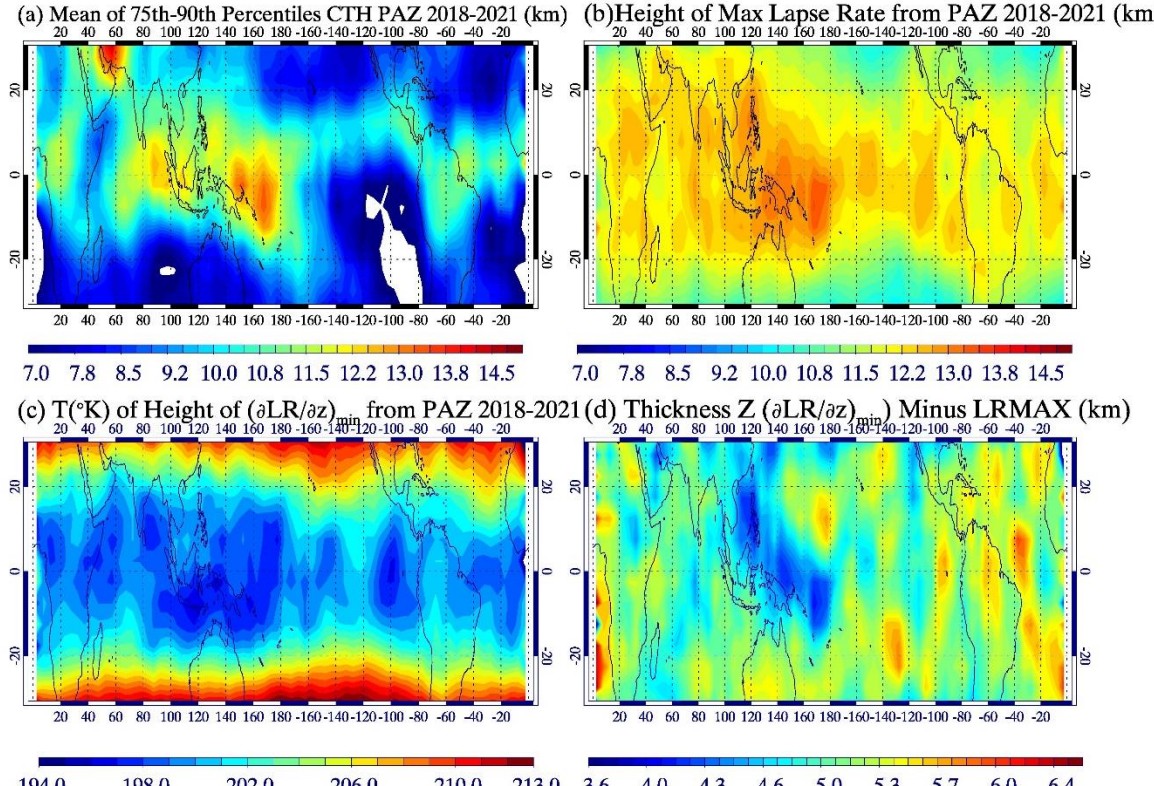

**Figure 7**: (a) Mean of 75th-90th percentiles of topmost height in each 5°x5° bin in which Δϕ>1mm from 2018-2021. White areas designate heights below 7 km. (b) Mean LRMAX in each 5°x5° bin. (c) Temperature corresponding to the height of $(\partial LR/\partial z)_{min}$. (d) Height of $(\partial LR/\partial z)_{min}$ minus LRMAX, which represents the estimated TTL thickness, in km. Note: The color scales are the same in (a) and (b) for comparison. CTOP < 7 km are depicted by white.

### 3.6 Sensitivity assessment of different thresholds for CTOP in the tropics and the relationships with LRMAX to glean new insights

To further corroborate the importance of LRMAX acting as either a cap or proxy for deep convection, we average instantaneous CTOP and LRMAX for heavily precipitating clouds into large spatial bins over the low-latitudes in 20° longitude bins latitudinally averaged between (20°S-20°N). Heavy rain is defined from individual profiles in which $[\Delta\phi]_{max}$>10 mm, in which the brackets represent looking for the value within the vertical column. We present this in Fig. 8 as a sensitivity analysis in which we estimate cloud tops based on different minimum thresholds of Δϕ. Mean binned CTOP is about 2.7 km below mean LRMAX when a Δϕ>1mm threshold is used, though with nearly a one-to-one spatial relationship. This LRMAX versus CTOP slope holds for Δϕ>0.8 mm with modest improvement in LRMAX minus CTOP (1.8 km), whereas reducing the CTOP-Δϕ threshold further comes at the expense of reduced spatial correlation between CTOP and LRMAX; this suggests that a lower threshold may be partially picking up tenuous signals less distinguishable from the background noise of Δϕ, and suggests that an acceptable Δϕ-threshold may be 0.8 mm to detect cloud tops.





### 3.7 Global and regional assessment of CTOP-LRMAX offset versus $[\Delta\phi]_{max}$

We synthesize our analysis of CTOP-LRMAX co-relationships by switching from absolute CTOP to CTOP relative to

LRMAX, CTOP minus LRMAX (henceforth CTOP-LRMAX offset), which not only helps normalize comparisons across different regions and the globe, but also sheds additional insight about the importance of LRMAX as a metric in which upper-level clouds are coupled, and ultimately how relative CTOP is more directly related to precipitation.

To accomplish this, we construct a global histogram of CTOP-LRMAX offset versus $[\Delta\phi]_{max}$ by choosing CTOP-LRMAX bin sizes of 0.8 km, and semi-log bins for $[\Delta\phi]_{max}$ with the following bin boundaries: [1.,1.5,2.25,3.25,4.5,6.,8.,12.,20.,40.,80.].

We note here that $[\Delta\phi]_{max}$ might be reminiscent of cloud optical thickness, $\tau$, in which cloud top temperature/$\tau$ bins were constructed using MODIS satellite data over the Tropical West Pacific (WP) and Tropical East Pacific (EP) in Kubar et al. (2007); we do not relate $[\Delta\phi]_{max}$ to $\tau$ here but note that such a comparison may be fruitful.

For high $[\Delta\phi]_{max}$ strongly precipitating clouds, strong cloud top radiative cooling may be driving/forcing the level of the maximum LR, such that we would expect CTOP and LRMAX to most closely coincide with each other. We also constructed

an analogous joint CTOP-LRMAX/$[\Delta\phi]_{max}$ for brightness temperature CTOP from IMERG/GPM, which we show in the Appendix (Fig. A1b), which qualitatively looks similar to the histogram from polarimetric-RO data in Figure 9a. There is a small population of PRO profiles which hint at clouds either weakly precipitating or non-precipitating that extend up to or just above 10 km above LRMAX; the cutoff for cloud tops from IMERG/GPM instead is 8 km. Whether or not those suggested polarimetric-RO very high clouds represent strong signals may require additional research, but cumulatively this is a very

small portion of cloud tops around the globe (<1%).

The cumulative distribution as a function of CTOP-LRMAX offset is also shown for Paz using both the 1 mm and 0.8 mm thresholds for CTOP, as well as the cumulative distribution for IMERG/GPM. This reveals that CTOP as ascertained by a 0.8 mm threshold from polarimetric RO statistically more closely resembles IMERG/GPM up to about a CTOP-LRMAX offset of about 0.0 km (~80% of global detected clouds), with Paz CTOP just a little shallower than IMERG/GPM. There's a top-

heavier component from PAZ with the 0.8 mm threshold well above LRMAX. Not surprisingly, the cumulative distribution of CTOP-LRMAX offset is shallower for the more stringent 1.0 mm $\Delta\phi$-CTOP threshold by about 1.3 km, which is a global value and is very close to the differences stated in the tropics for heavily precipitating clouds of 1.4 km.

For weakly or non-precipitating clouds, CTOP tends instead to be shallower and several km below LRMAX, with a relative maximum frequency between about 2-6 km below LRMAX. This may represent mid-level or even some of the deeper PBL

cloud tops; again, a regional analysis may be more illustrative, but these clouds are contributing less to total precipitation than cloud tops closer to LRMAX, which we'll discuss more momentarily. This is also the case for brightness-temperature CTOP, with a mode of cloud top heights about 1-2 km higher for modest or small values of $[\Delta\phi]_{max}$ for IMERG/GPM versus PAZ.

On Fig. 9a, we also show the 50th, 95th, and 99th percentile of $[\Delta\phi]_{max}$ for each of the height bins, which illustrates that the most heavily precipitating clouds tend to be near or even just above LRMAX; this is especially the case for the 95th or 99th

percentiles of $[\Delta\phi]_{max}$. The CTOP-LRMAX offset is 1.2 km and 0.4 km for the 95th and 99th percentiles of $[\Delta\phi]_{max}$, respectively,



and suggests that the greatest instability is near or just *below* cloud top for the most heavily precipitating clouds. For more weakly precipitating clouds, on the other hand, in which updrafts may be much weaker, they tend to reach levels below or far below LRMAX; LRMAX may thus be remotely set regionally in the most actively precipitating clouds.

Though we do not show the full histograms for the Tropical West Pacific (WP: 120°E-160°E; 5°N-15°) or the Tropical East Pacific (EP: 150°W-100°W; 5°N-15°N), we do present the 99$^{th}$ percentile of $[\Delta\phi]_{max}$ (z) for each region; the heaviest precipitation of 19.9 mm in the WP is associated with CTOP 0.4 km below LRMAX, while for the EP there's a peak about 1.2 km below LRMAX and then another one of 21.4 mm at 2.0 km above LRMAX. This is consistent with Kubar and Hartmann (2008) who used joint CloudSat cloud radar and AMSR-E rain rate retrievals to compare the vertical profiles of precipitating clouds over the WP and EP, and demonstrated that for a given precipitating cloud top height, EP rain rates were higher.

In Fig. 9b, we plot the distribution of the contribution of CTOP-LRMAX offset to the 95$^{th}$ percentile of $\Delta\phi_{max}(z)$, somewhat analogous to Figure 3c of Kubar and Hartmann (2008) in which they showed the probability distribution function of CloudSat cloud top height scaled by the corresponding rain rate for each height bin. Here, we multiply CTOP-LRMAX offset at a given height by the 95$^{th}$ percentile of $\Delta\phi_{max}(z)$, which we also refer to as the polarimetric phase difference ($[\Delta\phi]_{max}$) density. For PAZ, we show the global curve and the |LAT|<60° only curve for a CTOP threshold of $\Delta\phi$>1 mm, both with primary peaks
about 3-4 km below LRMAX, though with a secondary peak closer to LRMAX for the |LAT|<60° curve. We show the IMERG/GPM curve (which is fundamentally constrained for |LAT|<60°) as well, with the greatest contribution to $\Delta\phi(z)_{max}$ when CTOP is 0.4 km below LRMAX. We also examine the WP and EP with polarimetric RO data with the peak contribution over the WP 3.4 km below LRMAX and 4.2 km below LRMAX for the EP.

The area under each of the curves in Fig. 9b also corresponds to the total mean $[\Delta\phi]_{max}$, or precipitation; the corresponding
amplitudes of the WP and EP are appreciably larger than the global average, as the WP and EP are deep convective regions. The mean 95$^{th}$ percentile of $[\Delta\phi]_{max}$ for the globe from PAZ is 6.50 mm, 7.38 mm between 60°S-60°N, 9.92 mm for the WP, and 10.15 mm for the EP.

We also note that over the EP, there is a tendency of a deeper CTOP mode, about 2.2 km above LRMAX, to contribute to heavy precipitation rates, which would be in the lower to middle TTL layer; it may be slightly more likely for clouds to
penetrate LRMAX over the EP as LRMAX there is modestly shallower by about 0.5 km in the mean. We have insufficient samples to date to assess statistical robustness, however, it is notable in general that such an analysis can be used to estimate the fraction of clouds in an area in the TTL; as more PRO profiles continue to be collected, this can be done both over time and over smaller spatial regions.

We conclude in Fig. 10 with a global (PAZ) or nearly global (60°S-60°N; IMERG/GPM) assessment of the relationships
between CTOP-LRMAX offset versus nine bins of $(\Delta\phi)_{max}$. For clouds with the lowest $(\Delta\phi)_{max}$, or most lightly precipitating clouds, mean CTOP is anywhere from 4 km ($\Delta\phi$-0.8 mm threshold) to 6 km ($\Delta\phi$-1.0 mm threshold) below LRMAX, but the average of the most strongly precipitating clouds is ~1 km below LRMAX (IMERG/GPM) to 2.5 km below LRMAX ($\Delta\phi$-1.0 mm threshold); there is no statistically-significant difference between PAZ and IMERG/GPM for the two highest $(\Delta\phi)_{max}$ bins when a 0.8-mm threshold is used for PAZ, based on the overlap of the 99% confidence interval bars. There is also some overlap



between IMERG/GPM CTOP and PRO $\Delta\phi$-0.8mm CTOP for intermediate $\Delta\phi_{max}$. Between the two thresholds of $\Delta\phi$ (1.0 mm and 0.8 mm), there is only overlap of CTOP for the largest (mostly heavily precipitating) $\Delta\phi_{max}$ bin.

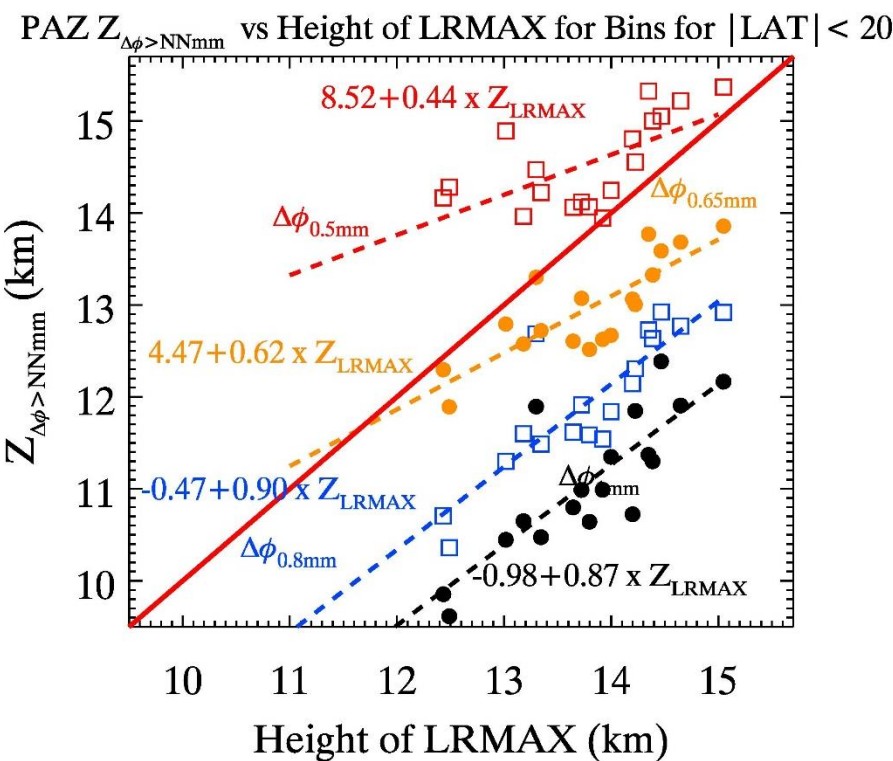

**Figure 8**: Uppermost height (between the surface and 20 km), inferred to be cloud top height, in which $\Delta\phi > 1$mm (black symbols and dashed line), $\Delta\phi > 0.8$ mm (blue symbols and line), $\Delta\phi > 0.65$ mm (orange symbols and line), and $\Delta\phi > 0.5$ mm (red symbols and dashed line) versus LRMAX for 20° longitude bins latitudinally-averaged between 20°S and 20°N bins. Red solid line is the one-to-one between CTH and $Z_{LRMAX}$. LRMAX is constrained to occur between 6 km and 20 km, and all profiles that are averaged are assumed to be heavily raining such that $[\Delta\phi]_{max} > 10$ mm. Also given for each is the best linear fit. The Pearson correlation coefficients are 0.81, 0.86, 0.78, and 0.66 for the thresholds of $\Delta\phi > 1$ mm, $\Delta\phi > 0.8$ mm, $\Delta\phi > 0.65$ mm, and $\Delta\phi > 0.5$ mm, respectively.






**Figure 9**: (a) Histogram of joint (CTOP-LRMAX) offset and $[\Delta\phi]_{max}$ from the Paz satellite. Vertical bin sizes are 0.8 km, and $[\Delta\phi]_{max}$ bins are semi-log with the following bin edges: [1.,1.5,2.25,3.25,4.5,6.,8.,12.,20.,40.,80.]. Also shown are various percentiles within each height bin of $[\Delta\phi]_{max}$, as well as the 99th percentiles of $[\Delta\phi]_{max}$ for the WP (120°E-160°E; 5°N-15°N) and EP (150°W-100°W; 5°N-15°N). The cumulative distributions for Paz and IMERG/GPM are also shown. (b) $[\Delta\phi]_{max}$-

Weighted CTOP to show the distribution of (CTOP-LRMAX) which comprise the 95th-percentile of $[\Delta\phi]_{max}$ either globally or given subregions.



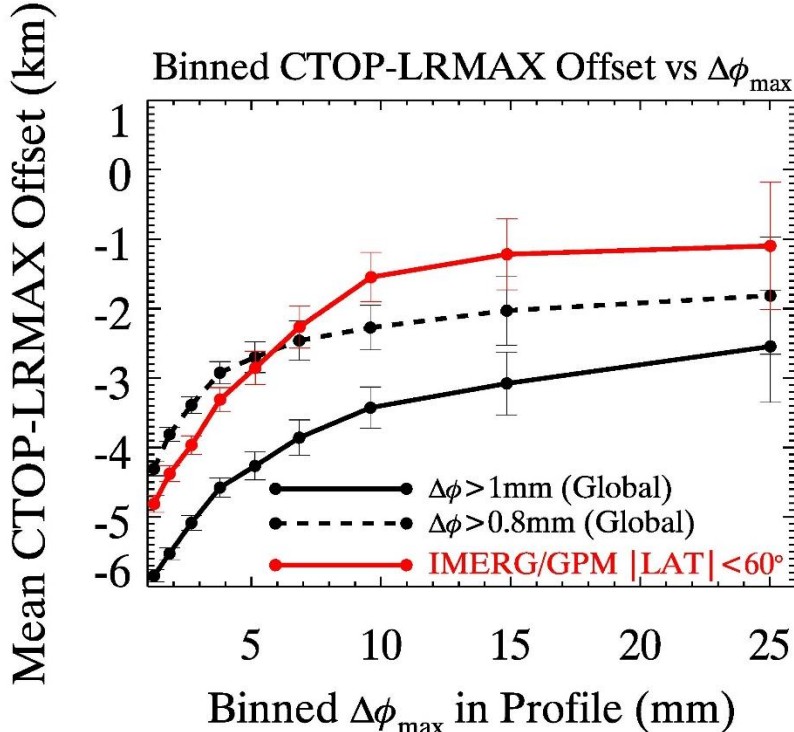

**Figure 10**: Mean CTOP-LRMAX offset versus binned $\Delta\phi_{max}$ for two different $\Delta\phi$-thresholds for CTOP from Paz of 1 mm and 0.8 mm, as well as IMERG/GPM brightness temperatures for |Latitude|<60°. Sampling error bars represent the 99% confidence interval, assuming normally distributed data, which is +/- $3 \cdot \sigma/\sqrt{N}$, in which $\sigma$ is the standard deviation within each set of pairs of $\Delta\phi_{max}$ categories and N is the sample size within each of the paired $\Delta\phi_{max}$ categories.

## 4 Discussion and Conclusions

In this study, we have explored the use of both conventional and polarimetric-RO profiles in characterizing key thermodynamic layers above the PBL. Our analyses focus on the upper troposphere, particularly the Tropical Tropopause Layer (TTL), and corresponding relationships to cloud top distributions, and provide estimates for how strongly such clouds may be precipitating. Beginning with a mid-latitude case study of a double tropopause case, with the two tropopauses separated by over 6 km, the case also demonstrates how stability decreases near the top of the layer between the first and second tropopauses, with stratosphere-like LRs just above the first tropopause, and then troposphere-like LRs just below the second tropopause (as similarly described as in Peevey et al. 2012). The upper tropopause in that case may be tropical in origin as the standard CPT resides at altitudes characteristic of the tropical tropopause, around 16-17 km, whereas the first tropopause is more characteristic of that found in mid-latitudes. However, in contrast to the tropics, in which a thinner TTL materializes as a result of very deep convection which more weakly affects the height of LRMIN, over the mid-latitudes a large separation between two tropopauses may often be indicative of a cold trough aloft, with air surging equatorward beneath the first tropopause



underneath poleward-moving air coincident with the tropical tropopause. Double tropopauses have been previously characterized before using GPS data (e.g. Randel et al., 2007) or the High Resolution Dynamics Limb Sounder (Peevey et al., 2012), but less so with PRO data and their ability to concurrently retrieve cloud profiles.

Next, we provided a suggestion for using the height of the minimum $\partial LR/z$, alternatively the sharpness-based CPT, as a robust marker of the top of the tropopause, and applied it globally. The standard CPT, while usually well-defined in the tropics
and often portions of the mid-latitudes, can often be located well into what is accepted as the lower-to-mid-stratosphere during the late autumn and winter seasons when there is limited, if any, sunlight. The height of $(\partial LR/\partial z)_{min}$ has the advantage of more closely coinciding with the tropical CPT for $|LAT|<25°$ than does the WMO tropopause, which is several hundred meters shallower, and the maximum sharpness of the negative vertical lapse rate gradient also closely tracks the WMO tropopause outside of the tropics, except for a small transition region between 25°S-30°S.

The results in Fig. 4 in the tropics are qualitatively similar to those of Xia et al. (2021) (their Tables 1 and 2) where the CPT is greater than the WMO LR tropopause (LRT), except that where the difference between the two tropopause methods is a minimum near the equator, CPT – LRT in Xia et al. (2021) is 1.8 km, in our analysis the differences are only 0.3 km in December and 0.42 km in June for $|LAT|<10°$, and are comparable to the CPT – LRT near the equator in Munchak and Pan (2014). The focus of Xia et al. (2021) is on the characterization of the tropopause by finding transitions of the refractivity
profile using the wavelet covariance transform (Gamage and Hagelberg, 1993), which they also compare against the LRT. Their refractivity-feature method of determining the tropopause is 0.56 km higher globally than the mean LRT. In our study, the global mean difference of the height of $(\partial LR/\partial z)_{min}$ versus the WMO tropopause is -0.03 km in June and +0.01 km in December, suggesting that the sharpness of the lapse rate gradient may have merit as a tropopause metric globally, and does not require the selection of an LR threshold (2°C/km) as is done for the WMO LR tropopause definition.

While the introduction of a sharpness-based CPT has such advantages, we also have presented a case in which the impacts of gravity waves, captured from a high-vertical resolution retrieval system such as RO, can complicate the calculation of either the standard CPT or the height of $(\partial LR/\partial z)_{min}$; relying on smoothed model data such as from NCEP also mischaracterizes the CPT by more than 1 km too high; future work may offer additional synergistic constraints between the WMO tropopause and the sharpness-based CPT for such cases.

One of the other main purposes of this study has been to utilize polarimetry to showcase the importance of these convectively or radiatively-driven tropospheric layers in the free troposphere to constrain and classify cloud layers. Using profiles from the Paz satellite launched in 2018, the polarimetric phase difference ($\Delta\phi$) of the vertically-oriented polarization from the horizontally-oriented polarization captures the vertical structure of ice and hydrometeors, especially heavy precipitation (Padullés et al., 2023), and therefore the corresponding cloud boundaries. We revealed that the seasonal
longitudinally-averaged LRMAX is just above the 75th percentile of cloud top heights at a given location (latitude), but at or just below the 90th percentile of cloud top heights. Moreover, globally, the 90th percentile of CTOP is somewhat more likely to be just above LRMAX in June globally than in December. These results suggest that LRMAX, which we consider the lower boundary of the TTL, is also effectively a boundary of most of the deeper clouds at a given location (latitude).



We next looked more generally at maps thus showing the horizontal distributions of the low-latitude relationships between
CTOP and different tropospheric layers including LRMAX. Some of the robust results that we report include:

  1)  The sharpness-based CPT defines the upper boundary of the troposphere is given by the height of $(\partial LR/\partial z)_{min}$, and
      the latitude at which this difference between the sharpness-based CPT and the standard CPT diverges may mark the
      edge of the tropics.

  2)  PRO data are used to test the hypothesis that the maximum lapse rate coincides with the tops of clouds, with the
strongest *a priori* association expected over the tropics. Instead, we find that the global LRMAX is coherent with the
      most heavily precipitating clouds anywhere, the latter of which is inferred by the strength of $[\Delta\phi]_{max}$. This CTOP-
      LRMAX relationship may be strongest for the most heavily precipitating clouds because of the high optical thickness
      of said clouds and their strong cloud top radiative cooling, significantly lowering the temperature relative to the
      ambient environment.

3)  While LRMAX is a good proxy for the TTL base, LRMIN represents a strong cap above which clouds are unlikely,
      except for possible exceptions in which LRMIN simply represents a strong inversion in the troposphere rather than
      tropopause top.

  4)  Over the tropics (20°S-20°N), binned CTOP is strongly correlated with LRMAX for profiles screened to be heavily
      precipitating based on $[\Delta\phi_{max}]>10mm$, although CTOP is on mean about 2.7 km below LRMAX when CTOP is
discerned by the uppermost height at which $\Delta\phi>1.0mm$. Reducing the CTOP-$\Delta\phi$ threshold to 0.8 mm increases
      heavily precipitating CTOP by 1.8 km while maintaining a nearly one-to-one CTOP-LRMAX slope (0.90), but
      decreasing the CTOP-$\Delta\phi$ threshold further to 0.65 mm decreases the CTOP-LRMAX slope to 0.62 since the signal-
      to-noise ratio is reduced.

  5)  Histograms of Joint Cloud Top Height (CTOP) and $[\Delta\phi]_{max}$ (proxy for precipitation) show that the heaviest raining
clouds are most likely to reach LRMAX or just exceed it, based on either PAZ or IMERG/GPM profiles, whereas
      weakly precipitating or non-precipitating clouds tend to be well below LRMAX (mean of 5.8 km below LRMAX for
      Paz CTOP with $\Delta\phi>1mm$ and mean of 4.8 km below LRMAX for IMERG/GPM for $[\Delta\phi]_{max} \sim 1.25$ mm).

  6)  We note that $[\Delta\phi]_{max}$ may be reminiscent of cloud optical thickness, $\tau$, in which cloud top temperature/$\tau$ bins were
      constructed using MODIS satellite data over the Tropical West Pacific (WP) and Tropical East Pacific (EP) in Kubar
et al. (2007), and that comparisons across different regions of relative CTOP/$[\Delta\phi]_{max}$ may be fruitful in future work
      as collection of additional polarimetric-RO profiles allow for more observations across smaller regions.

  7)  That the PAZ-constructed histograms qualitatively resemble the IMERG/GPM-constructed histograms suggests that
      the cloud top detection from polarimetric-RO holds promise for of using PRO for more systematic cloud top height
      and even cloud top height variability studies.

We end by commenting on the potential continued and growing importance of polarimetric-RO profiles of CTOP and co-
located thermodynamic profiles, especially in light of the expansion of these data in the commercial and public sectors.
Ongoing evaluation of the tropical tropopause layer processes, the evolution of the TTL with climate, and thus how investment



in PRO will enable continued essential research for the clouds and climate variability communities, may be instrumental in filling in gaps as the end of legacy missions (e.g. over two decades of continuous Terra and Aqua satellite data) draws near. In order to ensure both the robustness and utility of PRO as a resource of cloud profile and vertical precipitation information, future work validating how CTOP, LRMAX, and $[\Delta\phi]_{max}$ are related must continue, including the sensitivity of the CTOP $\Delta\phi$-threshold used.





**Appendix**




**Figure A**: Top: Joint histogram of relative CTOP/$[\Delta\phi]_{max}$, as in Figure 10a, from PAZ, using $\Delta\phi > 1$mm as the chosen threshold for CTOP, shown here for easy comparison with the joint histogram on the bottom constructed by using cloud top heights from



IMERG/GPM brightness temperatures between 60°S-60°N. For each relative CTOP bin, the 50th, 95th, and 99th percentiles of $[\Delta\phi]_{max}$ are shown in red for PAZ (top and bottom) and in black (bottom panel only) for IMERG/GPM.


**Code and Data availability.** Polarimetric RO profiles from the ROHP Experiment aboard the PAZ satellite and collocated key observables used in this study from GPM/IMERG are available as netcdf files at Jet Propulsion Laboratory's GENESIS repository at https://genesis.jpl.nasa.gov/ftp/paz_pol/cal_20200513/. Conventional dry RO profiles from Spire are available from NOAA through the following CDAAC site (https://data.cosmic.ucar.edu/gnss-ro/spire/noaa/nrt/level2/), specifically the
'atmPrf' profiles.

**Author contributions.** TLK and MTJ designed the study. TLK performed the calculations and the analysis with the assistance of MTJ. TLK made the figures and TLK and MTJ wrote the manuscript draft. TLK, MTJ, JK, and FJT discussed the research and worked on revising the manuscript.


**Competing interests.** The contact author has declared that none of the authors has any competing interests.

**Acknowledgements.** We thank Dr. Kuo-Nung Wang for generously preparing and providing the netcdf files containing the vertical profiles of $\Delta\phi$, temperature, and geolocation data as well as coincident IMERG/GPM brightness temperature cloud
top heights. We also thank Dr. Kuo-Nung Wang and Dr. Ramon Padullés for helpful discussion and feedback during the course of this study.

**Financial support.** The research described in this paper was carried out with support from National Aeronautics and Space Administration (NASA) ROSES Grant NH19ZDA001N-GNSS, under a contract with 80NM0018D0004, as well as NASA
Grant NNH22ZDA001N, Research Opportunities in Space and Earth Science (ROSES-2022), Program Element "A.44 Commercial Smallsat Data Scientific Analysis." Part of this research was carried out at the Jet Propulsion Laboratory, California Institute of Technology, under a contract with the National Aeronautics and Space Administration.

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
