# Peer review of "Characterization of Free Tropospheric Layers With Polar Radio Occultation Data"

_EGUsphere, 2024_

## Referee Comment (RC2)

Review of **Characterization of Free Tropospheric Layers With Polar Radio Occultation Data** by Kubar et al.

This paper presents an analysis of upper tropospheric and lower stratospheric structure using polarimetric radio occultation data, focusing on cloud-top heights (CTOP), lapse-rate-derived metrics (LRMAX, LRMIN), and the cold point tropopause (CPT). They find that LRMAX aligns closely with the tops of heavily precipitating tropical clouds and propose that it serves as a reliable proxy for the TTL base. Additionally, the authors introduce a "modified CPT" definition based on the sharpness of the lapse rate profile.

In its current form, the manuscript is not yet ready for publication, but I believe it has the potential to be a valuable contribution pending substantial revision. There are some worthwhile analyses throughout the study and have no doubt that polarimetric RO data is a valuable tool deserving of increased attention in the coming years. However, this paper spends a lot of its time focusing on tropopause definitions in ways I find to be problematic, and the overall purpose of the study gets lost in this.

General Comments:

1. In the title, "Polar" should really be "Polarimetric" in order to not be confused with the polar regions. Also, "Characterization of Free Tropospheric Layers" doesn't feel like it accurately describes most of the work presented in this paper.

2. While your introduction is a thorough literature review of relevant work, it is quite dense and difficult to follow at times, and it doesn't provide strong motivation for *why* you are performing this work. Why does this study matter? What is missing from the previous literature that you are working to rectify? I would suggest revamping this entire section to feel more cohesive and to ensure the motivation for your study is clearly defined.

3. Much of the focus of this paper (Sections 3.1 through 3.3) focuses on this "Modified CPT" and its use as a tropopause definition. It is unclear how this metric (minimum of the second derivative of temperature profile) represents a modification of the CPT, and it may be better framed as a novel tropopause definition altogether. Throughout these sections, you tend to compare this modified CPT with the CPT in scenarios where the CPT has no applicability. It is well known that the CPT decouples from the composition transition associated with the actual separation between troposphere and stratosphere outside of the tropics and therefore should not be used outside of the tropics as it has no real physical meaning. Therefore, any work showing how the modified CPT is better than the CPT outside of the tropics is not very compelling in my opinion. I strongly urge you to reconsider whether this new definition is necessary, and if other definitions may suit your purposes well

enough (WMO, PTGT, etc). All of this aside, I don't think these sections feel super relevant to the rest of the work looking at CTOP.

    a. Somewhat associated with this, I want to provide a note of caution to ensure that you are implementing the WMO definition correctly. Multiple times in the paper you describe the definition as being "the lowermost height at which an LR threshold of 2 C/km is sustained for at the least 2 km", while the exact definition from the WMO is "the lowest level at which the lapse rate decreases to 2 C/km or less, provided also the average lapse rate **between this level and all higher levels** within 2 km does not exceed 2C/km". I only mention this because the WMO definition is unfortunately frequently misapplied in published works.

4. I find the sections of the manuscript focused more on CTOP comparisons with temperature profile structure metrics to be more compelling, but this gets a bit lost due to the length and somewhat disorganized nature of the paper. Additionally, I think these sections would be even more compelling if it was more thoroughly motivated in the introduction.

5. The figures throughout the paper are quite messy and hard to digest, I would recommend thinking about different ways to show the wealth of information you are trying to convey in a more concise and digestible manner.

6. Throughout the paper, it is unclear how much of your analysis is focused upon the tropics versus the extratropics. For example, in the third paragraph of the introduction, you go from talking about the TTL, which is in the tropical regions by definition, to the tropopause inversion layer, which is a primarily extratropical feature. Please ensure that in your methods, analysis, and discussion that you are clear if you are focusing on the tropics, the extratropics, or both.

---

## Author Comment (AC1)

To summarize, this paper considers Earth radio occultation (RO) data obtained by standard RO and by polarimetric RO to explore quantities associated with clouds and the tropopause transition layer (TTL). First, in order to define the TTL vertical boundaries, they construct a new definition of the tropopause that can be used to bound the bottom side of the TTL. They found existing definitions inadequate for their purposes. They apply their definitions and explore the consequences for the TTL, the extent of double-tropopauses, and the depth of the troposphere. Then the authors explore relationships between cloud-top height according to various definitions of differential phase from polarimetric radio occultation (PRO) obtained by the rohp-PAZ mission. They find consistency of some of their results with previous findings by the first author.

I find this paper to be a stream-of-consciousness grab-bag of computations without well-defined purpose or intent, unorganized, abstruse in its presentation, with no interesting conclusion. It's as if the authors felt required to publish after an initial exploration of data without a coherent idea of why they should publish.

*First and foremost, thank you for your critical review; we appreciate your effort and dedication. As a quick overarching note, the primary thrust of the submitted paper was quantifying the tropospheric layering using PRO data, with a focus on the upper troposphere, the tropopause, and the extent to which the predominantly uppermost cloud tops are constrained by a global criterion that is valid nearly everywhere. Because we were after the universality of the importance of LRMAX as a measure of the TTL base in the tropics, as well as the extratropical equivalent – the Extratropical Transition Layer (ExTL) base (a layer we omitted previously but have added to indicate impartiality towards different latitudes – a critique of Reviewer Two that we were too heavily focused previously on the tropics), we now make sharper, albeit more generalized conclusions. The coherent through-line is the use of a general set of criteria to define the tropopause and the TTL/ExTL base throughout, layers driven by the minimum rate of change of lapse rate with height (tropopause) and the maximum LR, respectively. We address your concerns about not using a bottom-up approach to predict/categorize cloud top heights (e.g. the low-level equivalent potential temperature – namely the height in the upper troposphere in which the equivalent potential temperature, or nearly analogously the moist static energy, is the same as near PBL top) but rather the height coincident with where the longwave cooling by water vapor mixing ratio becomes insignificant and heating by $CO_2$, CO, and $O_3$ drive radiative heating, thereby modulating and decreasing the steep tropospheric lapse rates below which deep convection in the tropics and thick precipitating clouds in the extratropics are found.*

*The PULL mechanism of Kubar et al. (2007), which we cite in the study and discuss more below, applied the concept of the fundamental relationship of the precipitous decrease of longwave cooling rates in the upper troposphere, albeit over the tropics. The concept directly relates to the detrainment level/temperature of precipitating anvil clouds. Low-level equivalent potential temperature, on the other hand, can predict the optically thickest tropical clouds, but with less skill, and furthermore assumptions about entrainment profiles are usually needed to improve prediction of convective cores. LRMAX in our study works similarly to the convergence-weighted temperature in Kubar et al. (2007), but we apply it globally, and more simply, it requires no radiative transfer calculations. The novelty of using Polarimetric RO data is moving beyond quantifying the upper tropospheric layers in terms of thermodynamics (conventional RO profiles*

*only) and quantifying their coherence with the uppermost cloud top heights (available from the polarimetric enhancement), assuming those clouds are either precipitating or sufficiently thick.*

*Thus, the Polarimetric-RO thermodynamic/cloud monitoring synergy enables monitoring of variability and possible longer-term changes of statistics of frequency of overshooting/penetrating thick cloud tops into the TTL (Tropical Tropopause Layer) and ExTL (Extratropical Transition Layer).*

A couple of salient points. These are examples; fixing just these few points should not be construed as a response to this review. Much more serious repair is needed before it is resubmitted.

1- The title. The word "characterization" by itself suggests that the authors have no specific goal in mind. The tropopause transition layer is not widely regarded as the "free troposphere", as it is not free. The authors are using standard GNSS RO and "polarimetric" RO; "polar" RO suggests RO measurements obtained in the Earth's polar regions.

*Regarding the word "Polar" in the title, thank you for bringing this to our attention, and we agree with you – it turns out the title with the word "Polar" in the submitted manuscript was a mistake/truncation. It was supposed to be "Polarimetric" all along.*

*However, to your broader critique of the title, Reviewer 2 also thought that "Characterization of Free Tropospheric Layers" didn't really embody that well the salient points of the paper. As we point out in our responses throughout, we have substantially overhauled the paper and the title and abstract have been significantly modified to better capture the somewhat narrower and thus more focused topline goals and results of the study. Our new working title now is, "**Global Upper-Tropospheric Lapse Rate Constraints on Cloud Top Height Using Polarimetric Radio Occultation Data**."*

*Though the tropopause transition layer, usually better known as the tropical tropopause layer (TTL), no longer appears in our abstract, we address your comment here as there that the TTL "is not free." Generally, the tropopause itself has widely and historically been regarded as the top of the free troposphere; the TTL by definition is simply a layer of varying depth (which we do address and the stability through which is important for determining vertical mixing and the penetrative capacity of deep clouds) which is a vertical transition region from the (still free) troposphere to the stratosphere. Others have noted that this transition region is simply just that, a transition region separating the free troposphere below (thus below TTL base) from the stratosphere (above the TTL top), so one could make an argument, to your point, that the TTL itself is less free overall, more ambiguous, if you will. We do not go out of our way in the revised manuscript that the TTL region is "free." In fact, our current three statements about the free troposphere now are:*

*Lines 29-30*: *"The region immediately above the PBL, the free troposphere, is unstable to convection and less influenced by Earth's surface."*

*Lines 54-56*: *"The height of the maximum lapse rate height (henceforth LRMAX) in the free troposphere may set the effective height of maximum cloud detrainment below which temperatures are convectively adjusted (following Fueglistaler et al., 2009; Chae and Sherwood, 2010)."*

*Lines 169-170* (in reference to Fig. 1): *"The strong decrease of temperature with height, characteristic of the free troposphere, continues up to about 10.5 km."*

*Lines 485-486:* *"One of the other main purposes of this study has been to utilize polarimetry to showcase the importance of these convectively or radiatively-driven tropospheric layers in the free troposphere to constrain and classify cloud layers."*

*In literature that we already cite in the manuscript, but expand upon some more below, the TTL in some ways may be more characteristic of the free troposphere via examination of the ozone profile, which is a proxy for the troposphere by virtue of its relatively low values into the TTL up to the CPT. This suggests that while anvils associated with tropical convection may preferentially detrain near the TTL base, vertical mixing (and still relatively high LR air) permits PBL air to ascend to the CPT (e.g. see Figure 4 below of the Sunilkumar et al. 2017 paper which we also cite extensively in the manuscript):*

[Figure]

17 April 2014 - Trivandrum (17:30 IST)

*Still, even this description, to your point, does miss some of the subtleties of the structure; in Fueglistaler et al. (2009), the authors make the point that the vertical structure of ozone exhibits an S-shape, with slightly higher values in the mid-troposphere which then reach a local slight minimum at the convective detrainment layer near 200 hPa, with a sharp increase above 150 hPa. This thus might be more of a nod to your statement as the TTL not being free, hence it is a hybrid region, a transition zone as the name suggests.*

*Maybe a fuller nod to your statement though is that our primary argument is that LRMAX in the upper troposphere, along with convectively-generated detrained anvil clouds (in the tropics) or non-tropical thick precipitating clouds in the extratropics, are where they are because of the precipitous drop-off of water vapor above that height, as we make a more cogent effort of noting and documenting throughout. Such a statement is present in lines 414-418:*

> *"In the tropics, the drop-off in water vapor cooling profiles around 200 hPa is also the basis of the PULL mechanism in Kubar et. (2007), which is present well-away from deep convective areas and drives the level of upper-level clear-sky convergence, hence a*

*control on the detrainment height/temperature of convective anvil/cirrus clouds in deep convective areas. Above LRMAX, water vapor mixing ratios have been shown to observationally decline with height in areas of deep convection (e.g. Sunilkumar et al. 2017)."*

*We have similar statements to this earlier on in the manuscript as well about water vapor's rapid decline above upper-tropospheric convection on lines 55-56*

*Water vapor has a strong peak in net cooling at or just below 200 hPa, near the top of deep convection, whereas $CO_2$ warming above stabilizes the layer between the top of convection and the tropopause (Thuburn and Craig, 2002).*

*Or later on in lines 344-348 (not copied below), we also discuss how water vapor generally strongly decreases above LRMAX.*

*In the end, we do not drive home or explicitly state where the free troposphere ends because of some of nuances of the transition layer (TTL in the tropics and Extratropical Transition Layer, ExTL, in the extratropics) and the fact that it shares properties both with the free troposphere and the lower stratosphere. It is somewhat more of a moot point now as well given that we have revised the title so significantly. Our statement on line 30 about the free troposphere as the layer above the PBL extending up to the tropopause is a generic statement that can be found in many textbooks and references, many of which incorporate the free troposphere as extending to the tropopause.*

2- No motivation for using polarimetric RO is given. Stand-alone references do not suffice here inasmuch as PRO data are central to the authors' exploration.

*In the (original) submitted version of our manuscript, our introduction included a statement on lines 65-68 as to why using polarimetric RO is preferential for this work: "Understanding what drives the altitude of clouds and the CPT, the depth of the TTL, and their response to a changing climate make the high vertical resolution of all-weather RO observations and the sensitivity of polarimetric RO (PRO) to ice (Padullés et al., 2023) good tools for quantifying subtle changes in stability and layers of deep convective clouds."*

*However, to your point, and after significant reshuffling and reorganization of the Introduction, we agree with you that this motivation came perhaps too late, and now we motivate the use of PRO data in the first paragraph, on lines 32-35. In being responsive as well to Reviewer #2, we also include a better statement about the global nature of the observations used in the work:*

*Polarimetric RO (PRO) data add the sensitivity of ice to the high vertical resolution of all-weather RO observations (e.g., Padullés et al., 2023). This*

*enables analysis that quantifies subtle changes in thermal stability associated with deep convective clouds and, even more globally, non-convective raining clouds in the extratropical upper troposphere in a way not entirely possible or suitable with other passive or even active satellite sensors without external auxiliary data.*

*Then, on lines 66-71, we now have this paragraph, encapsulating both the physics that we are after and fundamental role of PRO data in achieving that:*

*"Because PRO profiles offer the opportunity to individually and systematically quantify the relationship between upper-tropospheric LRMAX and CTOP across all latitudes, understanding what drives the altitude of clouds, the global tropopause, the depth of the TTL/ExTL, and their response to a changing climate make the high vertical resolution of all-weather RO observations and the sensitivity of polarimetric RO (PRO) to ice (Padullés et al., 2023) good tools for quantifying subtle changes in stability and concurrent layers of deep convective clouds. It also makes the case for global criteria that enable proper identification of the transition between these vertical layers."*

3- Here is a typical indecipherable sentence (lines 424 – 427): "Though we do not show the full histograms for the Tropical West Pacific (WP: 120°E-160°E; 5°N-15°) or the Tropical East Pacific (EP: 150°W-100°W; 5°N-15°N), we do present the 99th percentile of for each region; the heaviest precipitation of 19.9 mm in the WP is associated with CTOP 0.4 km below LRMAX, while for the EP there's a peak about 1.2 km below LRMAX and then another one of 21.4 mm at 2.0 km above LRMAX." There are so many acronyms and sub-clauses that I cannot follow any of it; and this is the topic sentence of a paragraph. Almost every other sentence in the second half of this manuscript is like this.

*First and foremost, to your point and also to the points of Reviewer #2, the majority of our subanalysis examining smaller regions within the tropics (e.g. the West Pacific and East Pacific (WP and EP)) or in the mid-latitudes within the production of the histograms, has been removed, and this not only has simplified the histograms themselves, but also distilled the central messages of the two panels (PAZ PRO data on top and combined IMERG/GPM brightness temperature cloud top height with PAZ LRMAX retrievals on bottom). The regional analyses within the histograms shown in the submitted manuscript deserve a more thorough treatment, and we realized after your critical comment here and also of the other reviewer that the subject of the subtler differences within the tropics requires much more space and statistical significance testing outside of the scope of our study, and that such analysis is not appropriate here. Our submitted manuscript was already on the long side, so it made sense to remove these portions. The entire subsection of the description of the histograms has been significantly consolidated, hence there is no hint of the type of sentence that you include above in the revised manuscript. Any semblance elsewhere of any such writing has been removed or significantly revised.*

*To your larger point, the reorganization and simplification not only reduces the text necessary to describe this figure (and others), but we also have focused more on the*

*underlying physics and thermodynamics and we agree with you that excessive use of acronyms can dilute that message. We will point out though that Reviewer #2 thought that more merit was present in the latter part of the submitted version. But, we are critical of the previous version as you are, so substantial modification to the text was employed.*

*As mentioned, paragraphs like the one above have been completely removed or reworded. Since the reviewer did not point to other examples, we hope to have addressed them, but we can confidently state that we were our own strongest critics and have revamped many of these types of paragraphs into hopefully much more comprehensible ones.*

*The new histograms now no longer have lines/curves on them, and the x-axis has also been reformatted to be logarithmic, which provides more real estate to see the relationships of lightly to moderately precipitation clouds more clearly than before (in which this part of the Δϕ-max space was squashed).*

4- The authors show the TTL to extend up to 35 km in polar regions. This is not a serious finding.

*We reviewed the original text in case there was a typo somewhere, but we did not present this (such high-altitude TTLs) as a finding. A takeaway instead of the previously submitted version, which we emphasized in multiple locations of the text, was that the coldest temperature often is well into the lower or middle stratosphere away from the tropics; this was not a new finding but has been reported previously in the literature; the point of the original scatter plots of the high CPT altitudes was to showcase where and when this happens as a function of season (e.g. June versus December), and the asymmetry of the heights of the CPT between the two hemispheres. However, the new Figure 2 in the revised manuscript has been completely redone.*

*In the revamped Figure 2, which instead of scatter plots is a figure of the contours of lapse rates from ~6 km to 25 km, we more clearly highlight the layers of interest of our study, including how the uppermost clouds are entwined to regions of high LR. We feel that this is a more intuitive way of showcasing where the uppermost cloud top heights are with respect to the upper troposphere. One of the two tropopause metrics that we have developed, $(\partial LR/\partial z)_{min}$ is also shown on this figure. Though we no longer show it, it is evident where the absolute CPT is as discerned from the level corresponding to LR ~ 0°C km$^{-1}$. While this occurs above 20 km on average in many regions away from the tropics, as we note in the revised text, we stop looking for all tropopauses at 20 km when comparing the different metrics later on, such as in the new Figure 3.*

5- What physics is being tested here? What should we expect to find? I suspect even the simplest thought given to physics would have led the authors to consider an isentropic analysis and relationships to thermodynamic properties of the surface air, especially its equivalent potential temperature. Entropy/potential temperature makes no appearance in

this manuscript even though it is considered in one of the classic definitions of the tropopause, especially for studies related to clouds and convection such as this one.

*There are several ways of considering the drivers of the altitude/detrainment temperature of upper-tropospheric cloud, and the surface properties, including the saturation equivalent potential temperature at the LCL and its match to the level in the upper-troposphere, is behind the essence of the "PUSH" mechanism from Kubar et al. (2007) (hereafter KU2007). This is to address your point of consideration of "an isentropic analysis and relationships to thermodynamic properties of the surface air, especially its equivalent potential temperature." In KU2007, the PUSH mechanism indeed is that an air parcel will continue to ascend until it becomes negatively buoyant. However, this assumes that the parcel remains largely undilute, and calculations thereof showed some relationship with the thickest, rainiest tropical clouds, but not with the preponderance of anvil clouds, which are more fundamentally constrained by level at which clear-sky radiative cooling outside of convective regions declines precipitously and heating from $CO_2$ becomes important. This level is driven fundamentally by temperature via the Clausius-Clapeyron equation, whereas the near surface-based PUSH approach, to properly characterize the existence of clouds far below (20-30K warmer than), must assume entrainment profiles, far beyond the scope of what was done in KU07 or what we have done or wish to have employed here.*

*The physics of LRMAX are actually consistent with what KU07 defined as the convergence-weighted temperature; LRMAX can be much higher for a warmer or moister troposphere (or both); in general, lapse rates reach their maximum in the upper troposphere where saturated adiabats approach dry adiabats, and then LRs decrease as the influence of $CO_2$ becomes more important and emission from water vapor become negligible.*

*Some of the physics that we are describing above now appear in these two paragraphs, the fourth and fifth paragraphs of Sec. 3.6:*

> *For clouds with very high values of $[\Delta\phi]_{max} > 30$ mm, cloud tops and LRMAX exhibit the least variability with each other from either approach. In Fueglistaler et al. (2009), from a sounding climatology as part of the Southern Hemisphere Additional Ozonesondes (SHADOZ) program (at 7.5°S, 112.5°E), the observed lapse rate follows the moist adiabatic lapse rate up until about 250 hPa-200 hPa, and starts declining above this level. This is just below the level at which the moist adiabatic lapse rate reaches the dry adiabatic lapse rate (which happens around 200 hPa in the profile). This suggests a stabilizing constraint of this height encouraging the detrainment of upper-level convective anvil cloud; this is at work here in our results as well, but for both convective precipitating and non-convective thick extratropical precipitating clouds.*

> *In the tropics, the drop-off in water vapor cooling profiles around 200 hPa is also the basis of the PULL mechanism in Kubar et. (2007), which is present well-away from deep convective areas and drives the level of upper-level clear-sky convergence, hence a control on the detrainment height/temperature of convective anvil/cirrus clouds in deep*

*convective areas. Above LRMAX, water vapor mixing ratios have been shown to observationally decline with height in areas of deep convection (e.g. Sunilkumar et al. 2017).*

*It should be noted that the classic definition of the tropopause, which we use and compare as a reference against our tropopause metrics, simply utilizes lapse rate from the regular temperature profile, not the potential temperature profile, as the text now includes in lines 132-135:*

*The WMO definition of the tropopause, the lowermost height at which the LR decreases to 2°C/km or less, "provided that the average lapse rate between this level and all higher levels within 2 km does not exceed 2°C/km" (World Meteorological Society, 1957) usually captures the real tropopause, though occasionally, high latitude gravity waves (e.g. Figure A1) can lead to a mischaracterization of the tropopause height as in the lower stratosphere.*

*The potential temperature may be invoked for dynamical definitions of the tropopause, such as the potential temperature gradient tropopause (PTGT), normally in the realm of characterizing chemical species and their sharpest vertical gradients (e.g. our response to you above in #1 about ozone), but that also has a threshold (thus analogous to the WMO tropopause threshold requirement).*

*The review and works cited by Fueglistaler et al. (2009) (861 citations might make these authors' definitions reflective of the classical definition) mentions potential temperature to state: "We emphasize that there is no simple correspondence of, for example, boundary layer equivalent potential temperature and cold point potential temperature" and that for the TTL, potential temperature and temperature have equivalent behaviors: Examination of "…ozone profiles over Samoa (14°S) …" reveal … " that the sharp increase in mean ozone …" (starting) … "at 14 km coincided with decreases in the lapse rate of both temperature and equivalent potential."*

6- Lines 369 – 370: A plot (figure 7d) is presented for no reason at all.

*The original (submitted) Figure 7 focused on properties of the tropics, but that revised figure (the new Fig 5, lines 370-375) now covers most of the globe, and the Fig 5d in the revised manuscript now more clearly demarcates the tropics from the extratropics with regards to relatively large separation between the tropopause height and the LRMAX (TTL depth) over the tropics to maximum thickness away from deep convection (especially ~20°N/20°S) to much smaller separation between tropopause height and ExTL (Extratropical Tropopause Layer) base particularly over the high latitudes. The discussion of this panel is now included in lines 331-335 and lines 363-368:*

*Figure 5d shows the thickness of the height of $(\partial LR/\partial z)_{min}$ – LRMAX. Over the tropics, this thickness represents the TTL thickness, and it's greatest away from deep convection, particularly around 20°S/N over the Pacific in areas of weak compensating subsidence; it's also very large over the tropical Atlantic and portions of South America and Africa. Where deep convection is highest in altitude particularly over the maritime continent*

*and tropical western Pacific, the TTL thickness is smaller. Compared to the TTL thickness, over the extratropics the ExTL is much thinner, and is particularly thin over the north Pacific and Arctic Ocean regions.*

7- What can I learn from figure 9b? Is there anything interesting here?

*The original Figure 9b, which had the global versus subregional (tropical) curves and were intended to showcase the modes of CTOP-LRMAX, including the regional and global differences, were originally presented to demonstrate that the heaviest precipitating clouds had tops just below LRMAX (with some differences between IMERG, a passive satellite versus PAZ, our primary dataset in this study). However, visually, the revised (and simplified) histograms serve that purpose well enough, and the regional analysis, if it is done appropriately, deserves more attention and should instead be done in another study.*

*Furthermore, the old Figure 10 (and new Figure 8) serves the purpose more clearly of illustrating one topline result of the study; that even for the most heavily precipitating clouds, their tops are constrained by LRMAX (which is below the tropopause). We considered in the new iteration adding CTOP-Tropopause Height versus maximum $\Delta\phi$, but the range of the y-axis would have to be expanded too much, and thus we simply make a few statements about this in the revised manuscript.*

8- Instead of contriving a brand new definition for the tropopause—maximum derivative of lapse rate—which depends so heavily on a prior step of preprocessing, why not use an atmospheric reanalysis instead? Inasmuch as atmospheric analyses filter out internal gravity waves, which the authors also seek to do, why not at least try to use atmospheric analyses? This could have easily made this paper much more concise and focused.

*We sincerely appreciate your concerns about the amount of space devoted to the height corresponding with the minimum value of ($\partial LR/\partial z$), and whether it is even appropriate to devote this much effort to the various tropopause metrics. Indeed, Reviewer 2 was also concerned about the space devoted to the criteria associated with our tropopause metric, and we have significantly revised the manuscript to emphasize that we are after the criteria appropriate for high-vertical resolution observational data, such as RO profiles, which have been increasing in coverage and density in recent years, including from the private sector. We also address your comment about simply using analysis/reanalysis data, as well as the potential downsides of relying solely on those, as partially highlighted in our Figure 4 (old Figure 5, but updated which shows that ($\partial LR/\partial z$)$_{min}$ (from NASA CSDA Spire) and the WMO (UCAR Spire) tropopause agree with each other), but that the CPT from the model (NCEP) profile is too warm and nearly 1 km shallower than the CPT from Spire RO. While the model here in our example clearly has smoothed out any possible impact of gravity waves on corresponding vertical temperature perturbations, it is not evident that the smoothed profile here produces yields the correct CPT, or in this case, the model CPT is a little lower than the RO CPT, but if we look at the relative CPT (more below and in our revised manuscript), then the model relative CPT (with a sharpness component) would still be off.*

*As a clarification to your comment, we are not attempting to completely filter out the effects on temperature perturbations either in the TTL/ExTL or lower stratosphere by gravity waves, but rather have pointed out the challenges of said perturbations and have showcased robust tropopause metrics which pick out the stronger signal of the actual tropopause, which mostly supersedes effects from other phenomena (again, with the exceptions that we discuss). There can be a role in retaining such temperature anomalies that point to the stability structure or even to the presence in some cases of a double tropopause, the latter of which previously was a somewhat larger part of our study. While we have reduced our discussion overall of double tropopauses and have removed the figure of double tropopause frequency for brevity, coarser resolution models or reanalysis datasets are often unable to represent these temperature/stability structures; indeed Xian and Homeyer (2019) examined such vertical structure and characteristics in reanalysis datasets (ERA-Interim, JRA-55, MERRA-2, and CFSR) and showed that while the spatial patterns of double tropopauses were fairly consistent with radiosondes, double tropopause frequency was underestimated by up to 30%.*

*Thus, representation of the tropopause is highly dependent on vertical resolution, and RO data have often been used to evaluate biases of climate model data or different generations of reanalysis-derived tropopause statistics. While we don't do that here, our work can offer a template as such for evaluations of models. As an example of the sensitivity of tropopause/near tropopause characteristics to resolution, the previous generation of the ECMWF reanalysis products, ERA-Interim, is coarser in resolution than the current generation (ERA5), and either completely misses the temperature fluctuations associated with gravity waves just above the tropical tropopause or represents them as muted (+/- 0.5K) in amplitude compared to ERA5, in which the T-perturbations are on other order of +/-2 K (Hoffman and Spang, 2022: "An assessment of tropopause characteristics of the ERA5 and ERA-Interim meteorological reanalyses, ACP, https://acp.copernicus.org/articles/22/4019/2022/).*

**Not only can gravity waves be potentially muted or missed by models or reanalysis datasets, but more broadly, models may often miss thermal inversions above the cold point tropopause or the relative cold point tropopause (with sharpness requirement) that we introduce as an alternative just for Figure 3. A separate effort would be required to evaluate said tropopauses and inversions under a variety of conditions and locations, but nevertheless could be illustrative.**

*Okay, back to the changes that we have made - we have completely revamped the previous Figure 2, in light of your comments about lack of overall clarity of some of our figures, and you are indeed correct that it's important to consider whether it's worth our effort and a reader's time to distinguish the different tropopause definitions. Since lapse rate profiles are more at the heart of characterizing and thereby distinguishing the troposphere from the stratosphere, the new Figure 2 better illustrates these from the NASA CSDA Spire dataset for the same two seasons as before. This is also a cleaner figure overall than the scatter plots included in the originally submitted manuscript. In this way, our working tropopause metrics, in essence the*

*maximum absolute value of the sharpness of the vertical LR derivative, stands out and especially in the tropics, lies within the cold point (LR = 0K/km) and WMO contours (LR = 2K/km). Elsewhere, in the extratropics, the sharpness tropopause agrees closely with the WMO tropopause (Fig. 3). As a sidebar, hemispheric differences in the lower stratosphere are now more pronounced, a point we make, but less a primary finding since we are most concerned about the troposphere.*

*However, and this is important to our response to your above concerns, our goal is less of introducing a new definition of the tropopause but instead laying out criteria – the height corresponding to $(∂LR/∂z)_{min}$, that offers a physically consistent working metric of using observational data such as high-resolution RO to locate the tropopause. We also realize that the term "Modified CPT" was somewhat misleading, particularly since we only use the CPT as a constraint against possible outliers such that we only search for $(∂LR/∂z)_{min}$ between 6-20 km (rather than 25 km as in some cases previously) and only up to 1.1 km above the CPT, given the CPT is the height of the lowest temperature within this altitude range. We have removed most instances of "Modified CPT" throughout the text except once in the Introduction.*

*We also spend some space now stating that our goal is for our metric to be consistent with the WMO tropopause in a globally-averaged sense, however, there is additional processing that usually must be done to locate the WMO tropopause (to your next point), and furthermore, it is possible, especially during polar night, for even the WMO tropopause to not be found where the tropopause actually should be, at least for some outliers. This is because LRs may remain above 2 deg/km well into the lower stratosphere due to regions of relatively modest stability. We now briefly mention this in the updated manuscript as cases in which the WMO may misidentify the tropopause as too high, however, no set of criteria is perfect and there can sometimes can be some ambiguity as to where the real tropopause is. One concrete case of this is now shown in the Appendix (see figure included below).*

*Overall, we have tried to introduce this new criterion as a complement to the existing LR-based tropopauses such as the WMO, and our goal remains, and is more lucid now, of relating the vertical layering to the uppermost cloud top heights at all latitudes. We also choose the second derivative of temperature as an analog of sorts to work that the first author has been involved with in defining the top of the PBL, and overall our approach gets away from a threshold (WMO) approach.*

[Figure]

*Figure A. Example of a profile from PAZ (26 July 2018) in which none of the four tropopause metrics adequately describe the tropopause over polar night in the Southern Hemisphere at 72°S. (a) Temperature profile and CPT, (b) LR(z) and the WMO tropopause, and (c) ∂LR/∂z profile with the (∂LR/∂z)$_{min}$ indicated as well as the relative minimum temperature with the smallest (∂LR/∂z) (red X). All tropopauses are searched for between 6-20 km. The real troposphere likely is below about 7 km, as below that altitude higher LRs are observed.*

I very strongly urge a resubmission after the authors consider a wholesale reorganization of their research and presentation.

*Thank you again for your review, and your critical questions/comments, in conjunction with those of the other reviewer, prompted a thorough reinvestigation of both the objectives of our study and the distillation of key results. We believe that between the recommendations made in both reviews and our efforts towards overall simplification and consolidation where appropriate (ten original figures now down to eight, with nearly all of them modestly to significantly revised), a completely different title and substantially modified abstract, and refocused Introduction and simplified main body text (> 400 fewer words), that a wholesale reorganization has been accomplished for the better.*

**References in Our Responses Here:**

Chae, J. H., and Sherwood, S. C.: Insights into cloud-top height and dynamics from the seasonal cycle of cloud-top heights observed by MISR in the west Pacific region, J. Atmos., Sci., 67, 248-261, https://doi.org/10.1175/2009JAS3099.1, 2010.

Fueglistaler, S., Dessler, A. E., Dunkerton, T. J., Folkins, I., Fu, Q., and Mote, P. W.: Tropical tropopause layer, Rev. Geophys., 4, https://doi.org/10.1029/2008RG000267, 2009.

Hoffman, L., and Sprang, R.: An assessment of tropopause characteristics of the ERA5 and ERA-Interim meteorological analyses, Atmos. Chem. Phys., 22(6), 4019-4046, https://doi.org/10.5194/acp-22-4019-2022.

Kubar, T. L., Hartmann, D. L., and Wood, R.: Radiative and convective driving of tropical high clouds, J. Climate, 20, 5510-5526, https://doi.org/10.1175/2007jcli1628.1, 2007.

Padullés, R., Cardellach, E., and Turk, F. J.: On the global relationship between polarimetric radio occultation differential phase shift and ice water content. Atmos. Chem. Phys., **23**, 2199–2214, https://doi.org/10.5194/acp-23-2199-2023, 2023.

Sunilkumar, S. V., Muhsin, M., Venkat Ratnam, M., Parameswaran, K., Krishna Murthy, B. V., and Emmanuel, M.: Boundaries of tropical tropopause layer (TTL): A new perspective based on thermal and stability profiles, J. Geophys. Res.-Atmos., 122, 741-754, https://doi.org/10.1002/2016JD025217, 2017.

Thuburn, J. and Craig, G. C.: On the temperature structure of the tropical substratosphere, J. Geophys. Res.-Atmos., 107, https://doi.org/10.1029/2001JD000448, 2002.

World Meteorological Organization: Definition of the tropopause, Bulletin of the World Meteorological Organization, 6, 136-137, 1957.

Xian, T., and Homeyer, C. R.: Global tropopause altitudes in radiosondes and reanalyses, Atmos. Chem. Phys., 19(8), 5661-5678, https://doi.org/10.5194/acp-19-5661-2019.

---

## Author Comment (AC2)

Review of **Characterization of Free Tropospheric Layers With Polar Radio Occultation Data** by Kubar et al.

This paper presents an analysis of upper tropospheric and lower stratospheric structure using polarimetric radio occultation data, focusing on cloud-top heights (CTOP), lapse-rate-derived metrics (LRMAX, LRMIN), and the cold point tropopause (CPT). They find that LRMAX aligns closely with the tops of heavily precipitating tropical clouds and propose that it serves as a reliable proxy for the TTL base. Additionally, the authors introduce a "modified CPT" definition based on the sharpness of the lapse rate profile.

In its current form, the manuscript is not yet ready for publication, but I believe it has the potential to be a valuable contribution pending substantial revision. There are some worthwhile analyses throughout the study and have no doubt that polarimetric RO data is a valuable tool deserving of increased attention in the coming years. However, this paper spends a lot of its time focusing on tropopause definitions in ways I find to be problematic, and the overall purpose of the study gets lost in this.

*We very much appreciate the thoughtful and critical comments by the reviewer. We have spent considerable time implementing comprehensive modifications guided by your feedback and that of reviewer one. Just as a general note, our responses are in red Italics in this PDF.*

General Comments:
1. In the title, "Polar" should really be "Polarimetric" in order to not be confused with the polar regions. Also, "Characterization of Free Tropospheric Layers" doesn't feel like it accurately describes most of the work presented in this paper.

*Thank you for bringing this to our attention, and we agree with you – it turns out the title on the submitted manuscript was an oversight. It was supposed to be "Polarimetric" all along. To your other point, we have a significantly revised the working title to the following, which better emphasizes both the relationships and the constraints of upper-tropospheric lapse rates especially on upper-level cloud top heights: "Global Upper-Tropospheric Lapse Rate Constraints on Cloud Top Height Using Polarimetric Radio Occultation Data." While we continue to explore different criteria associated with the tropopause heights, part of our objective is to show that convection over the tropics, and the uppermost precipitating clouds over the extratropics, are more modulated/controlled by LRMAX rather than the tropopause itself. It is important in showcasing this to have a robust working definition of the tropopause, hence our effort spent on making sure we have a strong grasp of it.*

*We note as well that we have modified our description of the new criteria of the sharpness-based tropopause, and we now also include a relative cold point tropopause, which, as we state*

*in our abstract, is a "tropopause coincident to the smallest (∂LR/∂z) among cold points between 6 and 20 km." Outside of the tropics, the sharpness based tropopause $(\partial LR/\partial z)_{min}$ and the WMO tropopause agree with each other quite well, whereas the relative cold point (with sharpness component) tropopause is about 1 km or so higher than the WMO or $(\partial LR/\partial z)_{min}$.*

2. While your introduction is a thorough literature review of relevant work, it is quite dense and difficult to follow at times, and it doesn't provide strong motivation for *why* you are performing this work. Why does this study matter? What is missing from the previous literature that you are working to rectify? I would suggest revamping this entire section to feel more cohesive and to ensure the motivation for your study is clearly defined.

*Thank you, and after separating ourselves for some time from that section (the period between our original submission and receiving your review was > five months), we feel that being more objective is crucial to improving the readability of the Introduction, and you have brought up some excellent points/questions here and throughout your review. The other reviewer also critiqued that some of the exploratory analyses pursued in our study were at times too broad and not as coherent as they needed to be or as we had hoped they were.*

*Our original Introduction was 1363 words; we have significantly amended the Introduction which now contains about 132 fewer words (1231), but more importantly, it is more responsive to addressing the background more directly related to the topline topics/priorities addressed in the manuscript. We have included the tracked changes in one of the versions which does demonstrate that this portion of the paper was heavily edited. The clean version, with all of the edits accepted, has been submitted as well.*

*We want to make very clear as well that the Introduction has been substantially revised; some of the points made in the original submitted manuscript, particularly in the Introduction, were repetitive, and furthermore, we now draw a much stronger connection for why polarimetric RO profiles are essential to make the direct collocated comparison between thermodynamic properties and constraints of the upper troposphere, including the Tropical Tropopause Layer (TTL) and Extratropical Transition Layer (ExTL) and thick cloud tops as a function of inferred precipitation strength.*

*The core of why we are doing this work in part might be highlighted by this modified short paragraph (lines 67 – 72) - note as well that our revised Introduction and manuscript now includes the TTL and ExTL (Extratropical Transition Layer) to better represent the generality between the layering of the upper troposphere and thick precipitating cloud top height. We emphasize the last sentence for clarity just in our response to you here:*

*Because PRO profiles offer the opportunity to individually and systematically quantify the relationship between upper-tropospheric LRMAX and CTOP across all latitudes, understanding what drives the altitude of clouds, the global troposphere, the depth of the TTL/ExTL, and their response to a changing climate make the high vertical resolution of all-weather RO observations and the sensitivity of*

*polarimetric RO (PRO) to ice (Padullés et al., 2023) good tools for quantifying subtle changes in stability and concurrent layers of deep convective clouds. **It also makes the case for global criteria that enable proper identification of the transition between these vertical layers.***

*Before that paragraph, however, we do more clearly present now in the third paragraph of the Introduction, a gap in knowledge that we aim to close with our work:*

> ***Detection of the bottom and top of the substratosphere from observations, including the transition into the stratosphere with a single global criterion, have remained elusive.***

*To your excellent points throughout, the goal of this work is more global and general in nature and less tropics-focused. While we toggle back and forth at times between the tropics and extratropics, the objectives remain having a set of criteria in general.*

*To that point about treating the tropics and extratropics equally:*

> *An analog to the TTL exists in the extratropics, the Extratropical Transition Layer (ExTL), which also more clearly motivates the global approach to our analysis as a possible constraint of upper-tropospheric CTOP. Like the TTL, the ExTL, defined by the World Meteorological Organization (2003), is a region just below the thermal (LR tropopause), and is important since the square of the Brunt-Väisälä frequency, a surrogate of atmospheric stability, starts increasing ~2 km below the tropopause (Gettelman et al. 2011).*

3. Much of the focus of this paper (Sections 3.1 through 3.3) focuses on this "Modified CPT" and its use as a tropopause definition. It is unclear how this metric (minimum of the second derivative of temperature profile) represents a modification of the CPT, and it may be better framed as a novel tropopause definition altogether. Throughout these sections, you tend to compare this modified CPT with the CPT in scenarios where the CPT has no applicability. It is well known that the CPT decouples from the composition transition associated with the actual separation between troposphere and stratosphere outside of the tropics and therefore should not be used outside of the tropics as it has no real physical meaning. Therefore, any work showing how the modified CPT is better than the CPT outside of the tropics is not very compelling in my opinion. I strongly urge you to reconsider whether this new definition is necessary, and if other definitions may suit your purposes well enough (WMO, PTGT, etc). All of this aside, I don't think these sections feel super relevant to the rest of the work looking at CTOP.

*Thank you for bringing up this concern, and as we noted above, our goal is less of introducing a new definition of the tropopause but instead laying out a criterion – the height corresponding to $(\partial LR/\partial z)_{min}$, that offers a physically consistent working metric of using observational data such as high-resolution RO to locate the tropopause. We also realize that the term "Modified CPT" as it applied to what we were/are presenting was somewhat misleading, particularly since we have*

*slightly refined our search and only use the CPT as a constraint against possible outliers such that we only search for $(\partial LR/\partial z)_{min}$ between 6-20 km (rather than 25 km as in some cases previously) and only up to 1.1 km above the CPT, given the CPT is the height of the lowest temperature within this altitude range. We also spend some space now stating that we wish for our metric to be consistent with the WMO tropopause in a globally-averaged sense, however, there is additional processing that usually must be done to locate the WMO tropopause (to your next point), and furthermore, it is possible, especially during polar night, for even the WMO tropopause to not be found where the tropopause actually should be, at least for some outliers. This is because LRs may remain above 2 deg/km well into the lower stratosphere due to a lack of stratospheric ozone heating, due to gravity waves, or several factors. We now include an example in the revised Appendix of our manuscript (also shown below) where all four tropopause metrics are unable to capture the top of where the true tropospheric air is.*

[Figure]

*Figure A. Example of a profile from PAZ (26 July 2018) in which none of the four tropopause metrics adequately describe the tropopause over polar night in the Southern Hemisphere at 72°S. (a) Temperature profile and CPT, (b) LR(z) and the WMO tropopause, and (c) $\partial LR/\partial z$ profile with the $(\partial LR/\partial z)_{min}$ indicated as well as the relative minimum temperature with the smallest $(\partial LR/\partial z)$ (red X). All tropopauses are searched for between 6-20 km. The real troposphere likely is below about 7 km, as below that altitude higher LRs are observed.*

*In the revised manuscript, we have two working tropopause metrics and our goal remains using the discussion about tropopauses to characterize the top of the TTL/ExTL, which typically is separated from the tops of clouds, which are more constrained by the TTL/ExTL base. Part of the impetus of using the second derivative of temperature, or sharpness of the Lapse Rate, was to assess the tropopause as an analog of sorts to the PBL, which the first author of our study has used to define PBL top (inversion base) with conventional RO data (and using refractivity, N, rather than the lapse rate). It is encouraging that much of the time the WMO tropopause and $(\partial LR/\partial z)_{min}$ do agree with each other. During the revision process, in order to be true to more of a "modified CPT" (a term we no longer use in the paper), relative CPT criteria with sharpness component instead is introduced which gives credence to the fact that the CPT has physical meaning, but is not useful outside of the tropics; using our approach here shows that except for some outliers, it can be applied nearly anywhere. While we introduce a relative CPT here in terms of an additional metric (and with the constraint of a minimum $\partial LR/\partial z$), the general concept of a local cold point is not new, and has appeared in the literature before, particularly the extratropics:*

> *"In particular, high-resolution radiosonde measurements of the thermal and wind structure of the extratropical tropopause region exhibit a strong increase of temperature just above a sharp local cold-point tropopause." (Science Direct: Stratosphere/Troposphere Exchange and Structure - https://www.sciencedirect.com/topics/agricultural-and-biological-sciences/tropopause)*

*Finally, we have revamped Figure 2, which instead of scatter plots, now is a figure of the contours of LRs from ~6 km to 25 km; it more clearly highlights the layers of interest of our study, including how the uppermost clouds are entwined to regions of high LR. We feel that this is a more intuitive way of showcasing where the uppermost cloud top heights are with respect to the upper troposphere.*

a. Somewhat associated with this, I want to provide a note of caution to ensure that you are implementing the WMO definition correctly. Multiple times in the paper you describe the definition as being "the lowermost height at which an LR threshold of 2 C/km is sustained for at the least 2 km", while the exact definition from the WMO is "the lowest level at which the lapse rate decreases to 2 C/km or less, provided also the average lapse rate **between this level and all higher levels** within 2 km does not exceed 2C/km". I only mention this because the WMO definition is unfortunately frequently misapplied in published works.

*Thank you for bringing this up, and we completely agree that it's essential to not only be consistent with regards to the WMO definition but also to be thorough in its description. We have scaled back the use of the instances in figures in which the WMO definition is presented to Figure 1 (in which we calculate it ourselves) and in the now Figure 3 (old Figure 4). In the now Figure 3, the WMO tropopause comes directly from the Spire UCAR output, and indeed you are correct that the \*average lapse rate\* between the first level and a minimum of 2km the above*

*must have a lapse rate below 2C/km. On line 133, we also include a more precise definition of the WMO tropopause and include a portion of your statement in quotes (with a proper reference as well). Here is what we now write:*

> **The WMO definition of the tropopause, the lowermost height at which the LR decreases to 2°C/km or less, "provided that the average lapse rate between this level and all higher levels within 2 km does not exceed 2°C/km" (World Meteorological Society, 1957) usually captures the real tropopause, though occasionally, high latitude gravity waves (e.g. Figure A) can lead to a mischaracterization of the tropopause height as in the lower stratosphere.**

4. I find the sections of the manuscript focused more on CTOP comparisons with temperature profile structure metrics to be more compelling, but this gets a bit lost due to the length and somewhat disorganized nature of the paper. Additionally, I think these sections would be even more compelling if it was more thoroughly motivated in the introduction.

*Thank you. In fact, our paper now better highlights the LRMAX-CTOP relationships even more strongly, and in the Introduction, we include a more thorough primer about why this level, which might be defined by the TTL base in the tropics or the Extratropical Transition Layer (ExTL) base in the mid-latitudes, is important for the tops of tropical convective clouds or non-tropical thick precipitating extratropical clouds. While we retain some of the pre-existing literature review from before, we have consolidated where possible and sharpened the linkages between the importance between of three primary layers examined in the study – the high LR regions above the PBL to the TTL/ExTL base, where clouds are dominant, the decreasing LRs in the ExTL/TTL up to the tropopause, and then, to a lesser, the often present tropopause inversion layer (TIL) right above the tropopause, which as we now better characterize in the Introduction, has been observed at all latitudes. We have strengthened our description of this often ubiquitous inversion layer compared to the previous version.*

*But, more specifically, we address your primary concern earlier on, adding the following passage at the end of the first paragraph of the Introduction:*

*(Lines 32 – 35): "Polarimetric RO (PRO) data add the sensitivity to ice to the high vertical resolution of all-weather RO observations (e.g., Padullés et al., 2023). This enables analysis that quantifies subtle changes in thermal stability associated with deep convective clouds and, even more globally, non-convective raining clouds in the extratropical upper troposphere in a way not entirely possible or suitable with other passive or even active satellite sensors."*

*Later on, we have the following in our substantially updated portion of the Introduction, which in tandem addresses your appropriate concern about our previous hand-waving distinction between the tropics and extratropics (lines 67-71):*

> *Understanding what drives the altitude of clouds and the CPT, the depth of the TTL, and their response to a changing climate make the high vertical resolution of all-weather RO observations and the sensitivity of polarimetric RO (PRO) to ice (Padullés et al., 2023) good tools for*

*We noted above as well in part of our response to #2 that we introduce the ExTL as the extratropical analog (albeit imperfect) to the (tropical) TTL which motivates the metric employed, the height of the maximum lapse rate (LRMAX) to test as both a physical and statistical boundary for upper-level precipitating cloud tops.*

*In other spots (in the Introduction), we also tighten up the connection between the enhanced stability from the top of convection to the tropopause height, in a way that, to your point, was insufficiently clear before, while still invoking appropriate sources. Here is an example on lines 55-57:*

> *Water vapor has a strong peak in net cooling at or just below 200 hPa, near the top of deep convection, whereas $CO_2$ warming above stabilizes the layer between the top of convection and the tropopause (Thuburn and Craig, 2002).*

5. The figures throughout the paper are quite messy and hard to digest, I would recommend thinking about different ways to show the wealth of information you are trying to convey in a more concise and digestible manner.

*We appreciate this critique, and after reviewing the paper in its entirety with fresh, objective eyes, realize that too much information likely was trying to be packed into some of the figures, and many of them have been revamped or significantly updated. Others have been streamlined, or in many cases, simplified. In fact, only the original (and still existing) Figure 1 has not been revised at all! Here is a synthesis of some of the detailed changes that have been made:*

*Figure 2 now presents contours of LRs in the troposphere (and lower stratosphere) versus latitude as well as the mean height of the tropopause (using our sharpness-based criteria, $(\partial LR/\partial z)_{min}$), the 75th and 90th percentiles of CTOP, and mean LRMAX. This replaces the previous version which was a scatter plot, and might have been somewhat less clear (and for which there was a fair amount of criticism from the other reviewer).*

*The previous Figure 3 of the frequency of double tropopauses and the linkage with the meridional gradients of the height of min(dLR/dz) has been completely removed as double tropopauses are not a main thrust of our study, but more of a sidebar. However, when we checked the meridional gradients of the various tropopause metrics, including the newly introduced relative CPT (not shown), all of the results were broadly consistent with each other. That said, one question of the other reviewer had been why we hadn't just used reanalysis data to do much of the tropopause work, since the perception was that reanalysis data may filter out gravity waves. We still retain our Figure 1 which is an example of a double tropopause, but our primary response was that we have not explicitly attempted to removed the effects on temperature perturbations by gravity waves, but rather demonstrate that our tropopause*

*metrics are physically robust in picking out the tropopause signal for high-vertical resolution, potentially noisy temperature profiles.*

*The old Figure 6 has been removed, as it was showing some of the differences between PAZ and Spire in terms of their relationships with CTOP, but such an intercomparison really wasn't the objective of our paper, and furthermore, the CTOP/LRMAX geographic relationships are already included in the new Figure 2, so the old Figure 6 was somewhat redundant anyway.*

*The new Figure 5 now includes maps of all the parameters shown before, but includes the high latitudes to partly address your point about the lack of global representation in some of the figures. This also better highlights some of the points about the very cold tropopause over the tropics and the lower and warmer tropopauses over the extratropics, a point that has been made many times before but is clearly highlighted here, along with some of the asymmetries of the extratropics between the northern and southern hemispheres. This figure also contained some errors previously, and now should be much clearer.*

*The new Figure 6 (old Figure 8), is similar to before, but because of some slight smoothing that we employ using three-width boxcar smoothing, some former LRMAX outliers were removed, slightly improving the correlation between LRMAX and CTOP when the Δφ-0.8mm threshold is used; the 0.65 mm threshold for Δφ is also now acceptable, but only in the context of presumed heavily precipitating clouds. This is because for such clouds, the signal below is strong such that there is more confidence that the inferred cloud top is really connected to the primary cloud.*

*Finally, both you and the other reviewer thought that too much information was being shown previously on the histograms, and we have removed all of the sub-regional analysis curves, as there is not sufficient space to discuss the importance of those analyses, and furthermore, we agree that they took away from the overall message(s) of the histograms. The new CTOP minus LRMAX versus $\Delta\phi_{max}$ histograms are now presented for PAZ (top) and IMERG/PAZ (bottom), with nothing else overlaying. We also have changed the x-axis to be logarithmic which expands the part of the domain with non-raining or lightly precipitating profiles.*

*Instead of **10** main body figures as before, we now have **8** figures.*

6. Throughout the paper, it is unclear how much of your analysis is focused upon the tropics versus the extratropics. For example, in the third paragraph of the introduction, you go from talking about the TTL, which is in the tropical regions by definition, to the tropopause inversion layer, which is a primarily extratropical feature. Please ensure that in your methods, analysis, and discussion that you are clear if you are focusing on the tropics, the extratropics, or both.

*One of the first changes we made was removing the reference of the TTL in the abstract, and replacing it, where appropriate, with LRMAX, which is the universal metric with which we evaluate against CTOP using polarimetric RO data. This actually also ties in to your valid point more broadly about the apparent*

*disconnect of the previous version of portions of our Introduction and a fairly significant swath of our paper; our analysis really represents more of a near-global one, and the Introduction has been substantially rewritten. We admittedly were focused a little too much on the low-latitudes previously, and while fundamental concepts of the TTL motivated the some of the goals of this work, we have bridged the tropics and extratropics more comprehensively now with better references and contexts to analogs of the TTL in the extratropics – namely the Extratropical Tropopause Layer (ExTL), to showcase some of the similarities between the two layers and why defining and using LRMAX even outside of the tropics is still an appropriate constraint to test of upper-level tropospheric cloud tops.*

*We also alluded to it earlier in our responses, but we now include references and a description about how the Tropopause Inversion Layer is more of a global phenomenon than might be traditionally understood. But, in a nod to your point, the TIL in terms of nomenclature often is reserved for the extratropics, even though there is an equivalent layer in the tropics:*

*Lines 49-54:*

> *Though radiative-convective equilibrium calculations often fail to capture the frequent presence of a thermal inversion, it has been reported at all latitudes just above the tropopause (Birner et al., 2002; Noersomadi et al., 2019). $O_3$ radiative heating, or sudden stratospheric warming events in the extratropics (e.g. Zhang et al. 2019) can give rise to what is often referred to as the Tropopause Inversion Layer (TIL) (Birner et al., 2002; Randel et al., 2007). In the tropics, this layer of maximum stability immediately above the CPT is often a very strong inversion (e.g. Sunilkumar et al. 2017; Noersomadi et al., 2019). We refer to the altitude of this enhanced stable layer as the minimum lapse rate height, or in our study, the LRMIN.*

*References Used in Our Responses:*

*Birner, T., Dörnbrack, A., and Schumann, U.: How sharp is the tropopause at midlatitudes? Geophys. Res. Lett., 29, https://doi.org/10.1029/2002GL015142, 2002.*

*Gettelman, A. Hoor, P., Pan, L. L., Randel, W. J., Hegglin, M. I., and Birner, T.: The extratropical upper troposphere and lower stratosphere, Rev. Geophys., 49, https://doi.org/10.1029/2011RG000355, 2011.*

*Noersomadi, Tsuda, T., and Fujiwara, M.: Influence of ENSO and MJO on the zonal structure of tropical tropopause inversion layer using high-resolution temperature profiles retrieved from COSMIC GPS Radio Occultation, Atmos. Chem. Phys., 19(10), 6985-7000, https://doi.org/10.5194/acp-19-6985-2019, 2019.*

*Padullés, R., Cardellach, E., and Turk, F. J.: On the global relationship between polarimetric radio occultation differential phase shift and ice water content. Atmos. Chem. Phys., **23**, 2199–2214, https://doi.org/10.5194/acp-23-2199-2023, 2023.*

*Randel, W. J., Fei, W., and Piers, F.: The extratropical tropopause inversion layer: global observations with GPS data, and a radiative forcing mechanism, J. Atmos. Sci., 64, 4489-4496, https://doi.org/10.1175/2007jas2412.1, 2007.*

Sunilkumar, S. V., Muhsin, M., Venkat Ratnam, M., Parameswaran, K., Krishna Murthy, B. V., and Emmanuel, M.: Boundaries of tropical tropopause layer (TTL): A new perspective based on thermal and stability profiles, J. Geophys. Res.-Atmos., 122, 741-754, https://doi.org/10.1002/2016JD025217, 2017.

Thuburn, J. and Craig, G. C.: On the temperature structure of the tropical substratosphere, J. Geophys. Res.-Atmos., 107, https://doi.org/10.1029/2001JD000448, 2002.

World Meteorological Organization: Definition of the tropopause, Bulletin of the World Meteorological Organization, 6, 136-137, 1957.

Zhang, Y., Zhang, S., Huang, C., Huang, K., and Gong, Y.: The tropopause inversion layer interaction with the inertial gravity wave activities and its latitudinal variability, J. Geophys. Res.-Atmos., 124(14), 7512-7522, https://doi.org/10.1029/2019JD030309, 2019.